



# Above ground biomass dataset from SMOS L band vegetation optical depth and reference maps

Simon Boitard[1], Arnaud Mialon[1], Stéphane Mermoz[2], Nemesio J. Rodríguez-Fernández[1],
Philippe Richaume[1], Julio César Salazar Neira[1], Stéphane Tarot[3], and Yann H. Kerr[1]

[1]Centre d'Etudes Spatiales de la Biosphère, Université de Toulouse, CNES/CNRS/IRD/UPS, Toulouse, France
[2]GlobEO (Global Earth Observation), Toulouse, France
[3]IFREMER, BP 70, 29280 Plouzané, France

**Correspondence:** Arnaud Mialon (arnaud.mialon@univ-tlse3.fr) and Simon Boitard (simon.boitard2@univ-tlse3.fr)

**Abstract.** The Above Ground Biomass (AGB) is an essential component of the Earth carbon cycle. Yet, large uncertainties remain on its spatial distribution and temporal evolution. Improving the accuracy of the AGB estimates requires precise and regular monitoring. Satellite remote sensing offers such capabilities. In particular, the L-Band (1.41 GHz) Vegetation Optical Depth (VOD) derived from the SMOS (Soil Moisture and Ocean Salinity mission) multi-angle brightness temperatures is a good AGB proxy. Averaging the SMOS L-VOD over a year and linking it to a pre-existing AGB map is a well-established method to derive a spatial relationship between both quantities. After temporal extrapolation of this relation, global AGB time series are derived from the L-VOD, allowing to retrieve vegetation biomass values up to 300 Mg ha$^{-1}$ from 2011 onwards. This study focuses on this protocol to produce a harmonized AGB dataset from the L-VOD and analyses the impact of three factors on the AGB/VOD calibration. First, the influence of the orbit type (ascending or descending) on the estimation is quantified. Second, the relevance of using a single global spatial calibration or several regional ones is thoroughly discussed for the first time. Third, the AGB time series from this new dataset are compared against other published AGB time series to assess the validity of extrapolating a spatial relationship over time. These comparisons highlight that the produced dataset has more inter-annual variability than the other available time series and presents globally lower AGB estimates, particularly over the equatorial part of Africa. These two limitations are inherent to the input data and method used. Overall, the resulting AGB is coherent with the AGB map from the CCI Biomass version 4 and can be used in AGB studies. The freely accessible AGB dataset has been produced from the level 2 SMOS products, mixing ascending and descending orbits altogether and using a single global relationship between the AGB and the VOD. The spatial bias associated with the AGB estimates is also provided in the files. The AGB dataset is open access and the NetCDF files are available at: https://doi.org/10.12770/95f76ff0-5d89-430d-80db-95fbdd77f543 (Boitard et al., 2024).

## 1 Introduction

In the context of climate change and global warming caused by anthropogenic activities, understanding, monitoring and managing the Earth carbon cycle is critical (Grace, 2004). Vegetation biomass plays a key role in this cycle (Hese et al., 2005;





Houghton, 2005; Houghton et al., 2009). The forest biomass constitutes a large carbon reservoir as it contains most of the

carbon stored in the vegetation (Pan et al., 2013). Up to date the world's forests are a global carbon sink (Pan et al., 2011). Yet locally, they can act as either major sinks or prominent sources of atmospheric carbon dioxide (CO2) depending on land use practices, meteorological conditions or natural and anthropogenic wildfires (Clark, 2004; Wear and Coulston, 2015; Wigneron et al., 2020). Globally, the ability of forests to absorb carbon under a changing climate remains poorly known (Luyssaert et al., 2007). Hence, mapping the total Above Ground Biomass (AGB) and following its evolution in space and time is fundamental

to a precise and reliable monitoring of the Earth carbon balance.

  Estimating the AGB from the ground is challenging, as the densely vegetated areas are often remote (tropical and boreal forests) and hard to navigate through. In situ measurements are by definition local, and global monitoring would involve the installation and maintenance of a dense network of stations or multiple and frequent in situ measuring campaigns.

  This is why satellite remote sensing of AGB, with its global and regular coverage of the Earth surface, complements in situ

data very well. Optical indices have been used for decades to monitor terrestrial vegetation at medium and high resolutions (Zeng et al., 2022; Lu, 2006). Yet, such observations can not be relied on to estimate the AGB globally. They are mainly sensitive to the green component of the top canopy layer (Purevdorj et al., 1998) and tend to saturate over densely vegetated areas (Baret and Guyot, 1991). They are also heavily affected by atmospheric conditions such as clouds, aerosols or haze (Lu et al., 2017) which are common in moist tropical areas. For this reason, space-borne synthetic aperture radars (SAR), with their

all-weather capabilities, have proved useful to AGB estimation. Numerous studies, both theoretical and experimental, have demonstrated that SAR data respond to forest AGB up to a certain saturation point (Mitchard et al., 2009; Le Toan et al., 2011; Yu and Saatchi, 2016; Cartus and Santoro, 2019). This saturation point increases with wavelength (i.e., P- and L-band are more sensitive to AGB than C- and X-band). Beyond this level, the sensitivity diminishes. Interestingly, in case of dense forests, there might be a negative correlation observed between L-band backscatter and high AGB values (Mermoz et al., 2015).

More recently, the data acquired by space-borne passive microwave radiometers at L-band have received great attention for AGB estimation and related applications. These instruments deliver information complementary to optical indices and radar acquisitions. Despite their coarse spatial resolution (approximately 40 km), they benefit from a low emission contribution of the atmosphere. They can measure the microwave radiation emitted by the Earth surface in all weather and light conditions. The Soil Moisture and Ocean Salinity (SMOS) mission (Kerr et al., 2010) was launched in November 2009. Over land, the

SMOS passive microwave radiometer at L-band is used to retrieve the Surface Soil Moisture (SM) along with a vegetation optical depth (VOD). This radiative parameter characterizes the contribution of the vegetation layer to the measured signal. It simultaneously quantifies the opacity of the vegetation layer (how much it attenuates the signal from the underneath surface) and its own emission (how much the vegetation layer radiates). The L-band VOD (or L-VOD, (Wigneron et al., 2007)) is strongly correlated to the Vegetation Water Content (VWC) (Jackson and Schmugge, 1991; Grant et al., 2012). In addition,

the SMOS L-VOD is highly sensitive to AGB (Rodríguez-Fernández et al., 2018; Mialon et al., 2020; Frappart et al., 2020; Vittucci et al., 2019) and does not saturate as much as higher frequency band VOD (C, X or Ka) over dense forests (Rodríguez-Fernández et al., 2018; Chaparro et al., 2019; Wang et al., 2021). Consequently, as recently confirmed by Dou et al. (2023),

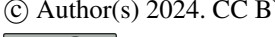



SMOS L-VOD is a good AGB proxy when averaged over a long enough time period. This time period typically covers a year to iron out the effects on L-VOD of diurnal and seasonal variations of the water content in the vegetation.

However, up to date, there is no equation nor model to directly compute the AGB from the VOD without using pre-existing AGB maps. Liu et al. (2015); Rodríguez-Fernández et al. (2018); Mialon et al. (2020); Wigneron et al. (2020) used several static AGB maps to derive spatial relationships between VOD datasets and reference AGB estimates. These static AGB maps have high spatial resolutions compared to SMOS, ranging from 100m (Santoro and Cartus, 2023) to 1000m (Avitabile et al., 2016) but are restrained to a year or a couple of years of reference. These maps may also present large discrepancies and
uncertainties (Mitchard et al., 2013).

To retrieve harmonized AGB time series from the L-VOD, the published literature focuses on establishing a spatial relationship that is later extrapolated over time. This study capitalizes on the same method as already described in Rodríguez-Fernández et al. (2018) and Mialon et al. (2020) and goes further by evaluating the impact of three factors on the VOD/AGB calibration. The work carefully investigates the influence of: using morning/afternoon overpasses separately or altogether, calibrating a sin-
gle global relationship or several local relationships and extrapolating a spatial relationship over time. Importantly, the derived AGB is provided with its associated error. The new AGB dataset estimated from 12 years of SMOS L-VOD is freely accessible from the SMOS french ground segment CATDS (Centre Aval de Traitement des Données SMOS).

Hereafter, Sect. 2 introduces the SMOS L-VOD products and the AGB reference maps. Section 3 describes the methodology. Section 4 presents the obtained results and a temporal analysis, further discussed in Sect. 5. Finally, the conclusions are
summarized in Sect. 6.

## 2   Data

This section presents the SMOS products used to estimate the AGB and the existing AGB maps that are used either to perform the AGB estimates or to evaluate our results.

### 2.1   SMOS products

SMOS (Kerr et al., 2010) is an Earth observation mission managed by the European Space Agency (ESA) and the Centre National d'Études Spatiales (CNES). This satellite, launched in November 2009, measures the Earth thermal emission as brightness temperatures (TB) at a frequency of 1.41GHz in full polarization, i.e. four stokes parameters, for a broad range of incidence angles (angles from 0 to 55° are used for the SM-VOD retrievals). SMOS crosses the equator at 06:00 am (06:00 pm) for ascending (descending) orbits.

The SMOS Level 2 (L2) products (Kerr et al., 2012) are used for this study. The core processing algorithm is based on the L-MEB radiative model (Wigneron et al., 2007) which uses the $\tau - \omega$ model (Mo et al., 1982) to account for the vegetation layer. In summary, on appropriate pixels, an initial couple (SM, VOD) is fed to a forward model computing the corresponding theoretical TB at all incidence angles and for both horizontal and vertical polarizations. The value of the (SM, VOD) couple is iteratively modified. The iteration process stops when the cost function derived from the sum of the squared weighted



differences between the modeled and measured TB reaches a minimum. The value of (SM, VOD) which minimizes the cost function is written out to the output product. The quality of the retrieval is evaluated by two variables reported in the L2 data. The first is the probability of Chi2 ($Chi2\_P$ field in the product) that measures how close the modeled TB are to the SMOS TB. The $Chi2\_P$ parameter ranges from 0 (poor quality) to 1 (excellent quality). The second one is the quality index labeled as $VOD_{DQX}$. It translates how the radiometric TB noise propagates through the retrieval model. This $VOD_{DQX}$ constitutes the

lower bound of the uncertainty on the VOD. Another point worth mentioning is that the L2 algorithm does not output VOD for footprints with little to no vegetation component such as the Saharan desert, Antarctica or any wet surfaces with no vegetation reported in the auxiliary datasets. The VOD is also not retrieved when the vegetation temperature is below 269 K as the frozen vegetation is transparent in L-band. The version 700 of the data is used in this study.

ESA freely distributes the L2 products (ESA, 2021) in a binary format. The data are provided on an Icosahedral Snyder

Equal Area (ISEA) 4H9 grid (Sahr et al., 2003) in swath mode with an almost constant inter grid points distance of 15 km. Considering the L2 file format and spatial projection, the products are pre-processed to work with common temporal and spatial grids for all input files. The SMOS orbit files are aggregated into daily ascending and descending maps. Where multiple measurements were acquired the same day, the data with the highest quality and closest to the sub-track is selected. These cases are mainly located under high (>60 ° N) latitudes. Finally, the variables are reprojected to the Equal-Area Scalable Earth

Grid (EASE-Grid) version 2.0 (Brodzik et al., 2012, 2014) Global, equal area projection (EPSG: 6933), with a grid sampling of 25 km at 30 degrees of latitude. The resampling is performed through a Delaunay Triangle interpolation if possible (3 valid interpolants exist), or linear if only 2 valid interpolants are available. The EASE 2 grid offers the advantage of being regular and is the default spatial projection for the SMOS CATDS ground segment products such as the Level 3 (L3, (Al Bitar et al., 2017)). An example of the 2018 averaged L2 v700 VOD values resampled to the EASE 2 Grid at 25 km mixing ascending and

descending orbits (AD) is displayed in Fig. 1.





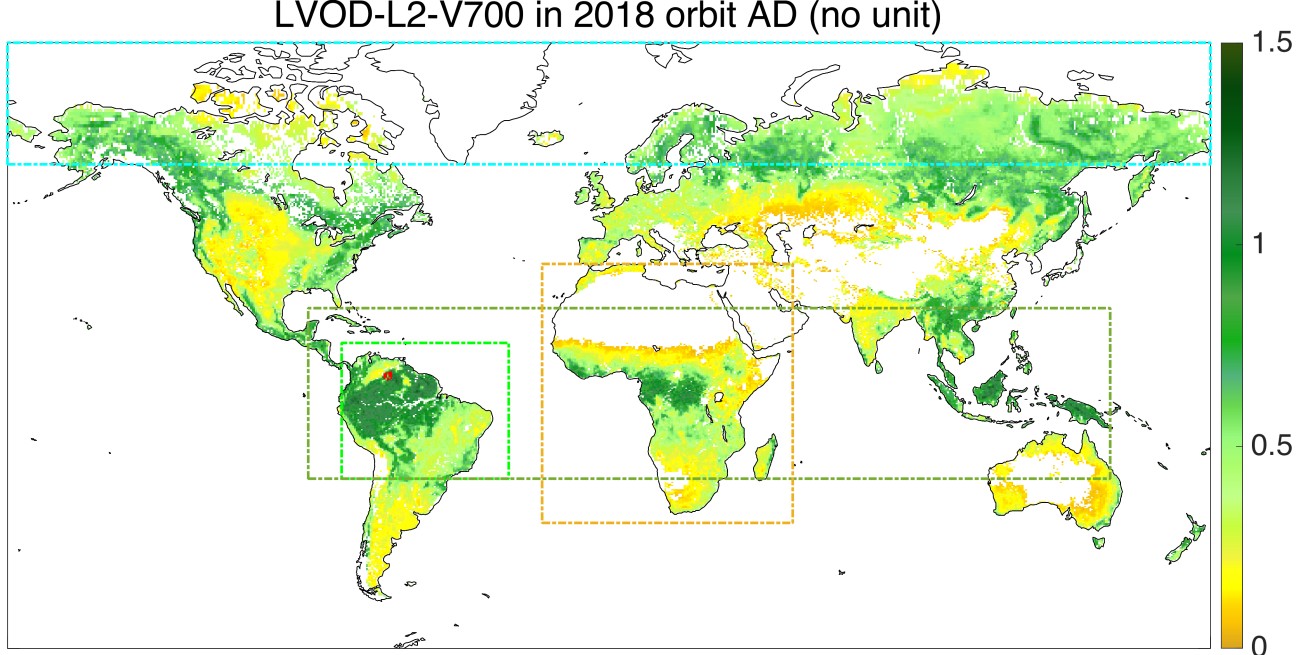

**Figure 1.** L-VOD from SMOS L2 v700 products averaged over 2018. Ascending and descending orbits are mixed. Values where RFI contamination is higher than 20 % or where the $Chi2\_P$ is lower than 5 % are filtered out. Pixels with an AGB reference value equals to 0 are also filtered out and outliers are discarded. The colored rectangles show the extent of the different regions considered in the study

## 2.2 Above Ground Biomass reference maps

This study involves the three AGB reference maps described below. The AGB map from Santoro and Cartus (2023) and Avitabile et al. (2016) were used as calibration data for the production of the SMOS-based AGB maps. The yearly AGB maps from Xu et al. (2021b), covering the years 2000-2019, were used to compare and contextualize the AGB time series resulting from this work. The AGB reference maps were averaged from their native sampling to the same EASE grid version 2 25 km as the interpolated L2 products.

### 2.2.1 ESA Biomass CCI 2017-2020 global AGB map

The ESA Climate Change Initiative (CCI) Biomass maps (Santoro and Cartus, 2023) are outputs of the ESA CCI Biomass project. The AGB estimation was derived using the radar backscatter intensity data captured by the Phased Array-type L-band Synthetic Aperture Radar (PALSAR2) on the Advanced Land Observing Satellite (ALOS2), and the Sentinel-1 satellites. The process of AGB estimation was also based on LIDAR metrics and surface reflectances. The AGB map is updated annually with a pixel size of 100x100m. Each CCI AGB map comes with its associated standard deviation which is a combination of the standard deviations from the input data, the modelling algorithms and the merging procedure. Version 4 (v4) of the maps is





used for this work. Global maps for the years 2010 and 2017-2020 are available in v4. Panel (a) of Fig. 2 shows the distribution

of AGB on Earth in 2018 according to Santoro and Cartus (2023) resampled to EASE 2 grid at 25 km.

### 2.2.2 Avitabile 2010 pantropical AGB map

The 1-km pixel size AGB map from Avitabile et al. (2016), limited to the pantropical region, was produced by integrating two pre-existing AGB maps from Saatchi et al. (2011) and Baccini et al. (2012) for the year 2010. This integration was achieved by employing an independent set of field observations and regionally adjusted high-resolution biomass maps, which were then

standardized and aggregated to produce almost 15,000 AGB reference data at a 1-km scale. The data fusion approach, involving bias correction and a weighted linear averaging technique, was implemented separately within distinct zones characterized by uniform error profiles in the Saatchi et al. (2011) and Baccini et al. (2012) AGB maps. The map from Avitabile et al. (2016), resampled onto the EASE Grid 2 at 25 km, is shown on panel (c) of Fig. 2. Panel (d) exhibits the differences of the AGB from Avitabile minus the one from CCI 2018 v4.

### 135 2.2.3 Xu 2000-2019 global AGB maps

Xu et al. (2021a) calculated the live (above and below ground) carbon biomass of global terrestrial ecosystems on an annual basis, covering the years 2000 to 2019. Reference data, including 100,000 plots coupled with airborne LIDAR data covering more than 1 Mha of tropical forests globally and satellite lidar survey detailing the height structure of global vegetation across over 8 million sample footprints, were used to feed a machine learning model. Reference data were converted into estimates of

both above and below-ground biomass (BGB) using established allometric models and root-to-shoot ratio. These data are used as training data for the machine learning algorithm, together with microwave and optical satellite imagery collected from 2000 to 2019. The annual carbon density maps of live woody vegetation are distributed under Xu et al. (2021b). The living biomass (Carbon density divided by 0.49 according to Xu et al. (2021a)) map for 2018 reprojected to the EASE 2 grid at 25 km appear on panel (e) of Fig. 2. Panel (f) shows the difference of the living biomass (AGB+BGB) from Xu minus the AGB from CCI v4

for the year 2018.

Earth System
Science
Data

**Figure 2.** Reference maps used in this study and the spatial differences between them. **(a)** Above ground biomass from Santoro and Cartus (2023), **(c)** Above ground biomass from Avitabile et al. (2016), **(d)** Above ground biomass difference between Avitabile et al. (2016) minus Santoro and Cartus (2023), **(e)** Above and below ground biomass (BGB) in 2018 from Xu et al. (2021a), **(f)** Biomass difference between Xu et al. (2021a) minus Santoro and Cartus (2023). All units are in Mg ha$^{-1}$ and all maps are resampled to the EASE 2 grid at 25 km

## 3 Methods

The methodology used to derive to AGB is detailed in Fig. 3. It exhibits the input and output datasets and the set up. The purpose of this workflow is to calibrate a relationship between the L-VOD and the AGB and quantify the impact of three factors on the calibration. The first factor is the influence of the local time of the SMOS overpasses. SMOS has a polar sun synchronous orbit. During the ascending orbits (around 6:00 am local time), thermal equilibrium is reached and the vegetation temperature is supposed to be close to the air temperature at 2m height (Kerr et al., 2012). This hypothesis is not valid for descending





orbits (6:00 pm) and may lead to a larger uncertainty in the VOD retrievals. The second factor is the impact of calibrating one global relationship versus several regional ones. The regions considered in this study appear in colored rectangles on Fig. 1: the amazon [25° S-15° N] and [80° W–30° W], the tropics [25° S-25° N] and [90° W-150° E], the African continent [37° S-37° N]

and [20° W-55° E] and the north region [50° N-90° N]. The third and last factor to be tested is the relevance of extrapolating a spatial relationship over time.

For conciseness, this paper describes the methodology and results with the ESA CCI Biomass 2018 map (CCI 2018, (Santoro and Cartus, 2023)) as the calibration dataset. Therefore, the relationship between these AGB reference values and the SMOS VOD estimates acquired the same year (2018) is described and discussed. Ultimately, all years of SMOS data and other AGB

reference maps were used for the final dataset production. Xu et al. (2021a) maps are only used for comparison.

**Figure 3.** Overview of the methodology for estimating yearly AGB maps from the SMOS L-VOD and an AGB reference map.



The workflow is divided into four steps numbered from 1 to 4 in Fig. 3. During step 1 (1- Pre-processing), the SMOS products and AGB calibration maps are aggregated, interpolated and averaged for the common period (here 2018 for CCI used as the reference) as described in Sect. 2. The L-VOD was aggregated for ascending and descending orbits separately to investigate
the influence of the orbit local time on the calibration.

In step 2 (2- Processing) the SMOS L-VOD daily measurements are masked, filtered, and temporally averaged. Only pixels over continental surfaces are considered, as provided in the 1km USGS (US Geological Survey) land-sea mask aggregated into the EASE 2 grid. Low-quality VOD are removed based on the $Chi2\_P$ and the level of Radio-Frequency Interferences (RFI).
These RFI are emitted by human-built equipment and mask the Earth natural emission over large areas. They prevent SMOS from globally covering the Earth (Oliva et al., 2016). The SMOS footprints with more than $20\%$ of SMOS TB contaminated by RFI or where $Chi2\_P$ is lower than $0.05$ are filtered out. The VOD temporal series of each footprint are also checked for potential outliers: the values outside an interval of two standard deviations around the yearly average are discarded. They are also checked for spurious discontinuities that are not induced by VOD or biomass changes. A small region of 85 EASE 2 grid
pixels in the north eastern part of the Amazon rain-forest requires a particular filtering. Over this area, identified in red in Fig. 1, the VOD times series experience a sudden jump in May 2015. This artifact is due to the 41r1 cycle update of the European Center for Medium-Range Weather Forecast (ECMWF) surface model which led to a discontinuity in the skin temperature (as used in SMOS auxiliary files, (Kerr et al., 2012)). This region is masked out for the analysis conducted in this study. No other affected areas have been identified.
After cleaning, the VOD time series are then temporally averaged on a yearly basis. This step is mandatory to iron out the effects on L-VOD of the diurnal and seasonal variations of the vegetation water content. At this stage, footprints with an AGB reference value equal to 0 are discarded, as they are not useful to the AGB estimation. The region mask is also applied to quantify the impact of calibrating several local relations versus one global relation. Figure 1 shows an example L2 VOD over the full globe for the year 2018 after the processing described above.


In step 3 (3- Fitting in Fig. 3), the AGB reference map is then compared pixel-wise against the annual SMOS L-VOD map (red points in the 3 - Fitting part of Fig. 3) for the same year (for example SMOS L-VOD in 2018 against ESA Biomass CCI AGB map for the year 2018) to check the relevance of a logistic relationship. Following the methodology described in Rodríguez-Fernández et al. (2018), the annual L-VOD is binned into 0.05-width bin. In each bin, the mean AGB from the
reference map is computed (black points in the $3 - Fitting$ part of Fig. 3) and the set of parameters of the logistic function that best fits the mean AGB/L-VOD distribution is estimated. This logistic function is defined in Eq. 1:

$$AGB = \frac{a}{1 + e^{-b(VOD-c)}} + d \tag{1}$$

where $a$, $b$, $c$ and $d$ are the free parameters. In equation 1, $AGB$ is in $Mgha^{-1}$ and the $L-VOD$ is dimensionless. Hence $a$





and $d$ are in $Mgha^{-1}$ and $b$ and $c$ have no dimension.

In step 4 (4- Validation in Fig. 3), the optimized logistic relationship is applied to the configured (particular orbit, regional or global coverage) VOD to produce AGB estimates. The resulting AGB is in turn evaluated against the calibration data. The AGB estimations are compared to the calibration values through 3 classical indicators: the Pearson's correlation coefficient

(R), the Root Mean Square Error (RMSE) and the mean difference (Bias). These statistics, complemented with maps and time series comparisons, lead to the selection of one optimal parametrization. This optimal spatial relationship is propagated to other annual VOD. The result is the AGB time series estimation for all SMOS years from a reference map (Output data at the bottom of Fig. 3).

Finally, the reliability of the AGB estimation is computed. Two quantifiable uncertainty sources were identified : (i) the uncertainties associated to the input data that are propagated through the process and (ii) the bias resulting from the logistic fit between the AGB and the SMOS VOD.

For i), The Monte-Carlo method is used to propagate the standard deviation of the reference AGB (available for the CCI but not for the other calibration maps). A dataset of $N = 10,000$ reference AGB maps is created. For each pixel, the AGB

value is extracted from the Gaussian distribution characterized by its mean being the reference AGB from the CCI, and its standard deviation being the CCI uncertainty. From these $N$ reference AGB maps, $N$ logistic fits and $N$ AGB estimations are performed. The standard deviation of the $N$ estimated AGB can then be computed per pixel. The same Monte Carlo method is applied to propagate the uncertainty associated to the yearly averaged VOD. For each footprint, the yearly $VOD_{DQX}$ is obtained through the quadratic mean of the daily $VOD_{DQX}$. This value is further divided by the square root of the number of

observations, as the input TB radiometric noises are considered independent from one day to the other. The yearly VOD and associated yearly $VOD_{DQX}$ are then used to create the dataset of $N$ VOD maps.

For (ii), the dispersion (*std*) of the estimated AGB for the reference year is derived against the input AGB values. The estimated AGB for the reference year is binned into 10 Mg ha$^{-1}$ bins. The mean of the input AGB values is computed within each bin (blue points on Fig. 4). The scattering of the estimated AGB values with respect to the input AGB map is computed per

bin as the gap between the 84 and 16 percentiles of the differences between the reference and estimated AGB. The result is a discrete spatial bias distribution of approximately 30 values (blue bars on Fig. 4). This distribution is propagated to other years. For each year, the bias map is built by dispatching to all pixels the reference bias value of the bin into which their estimated AGB values fall (see Fig. 4).

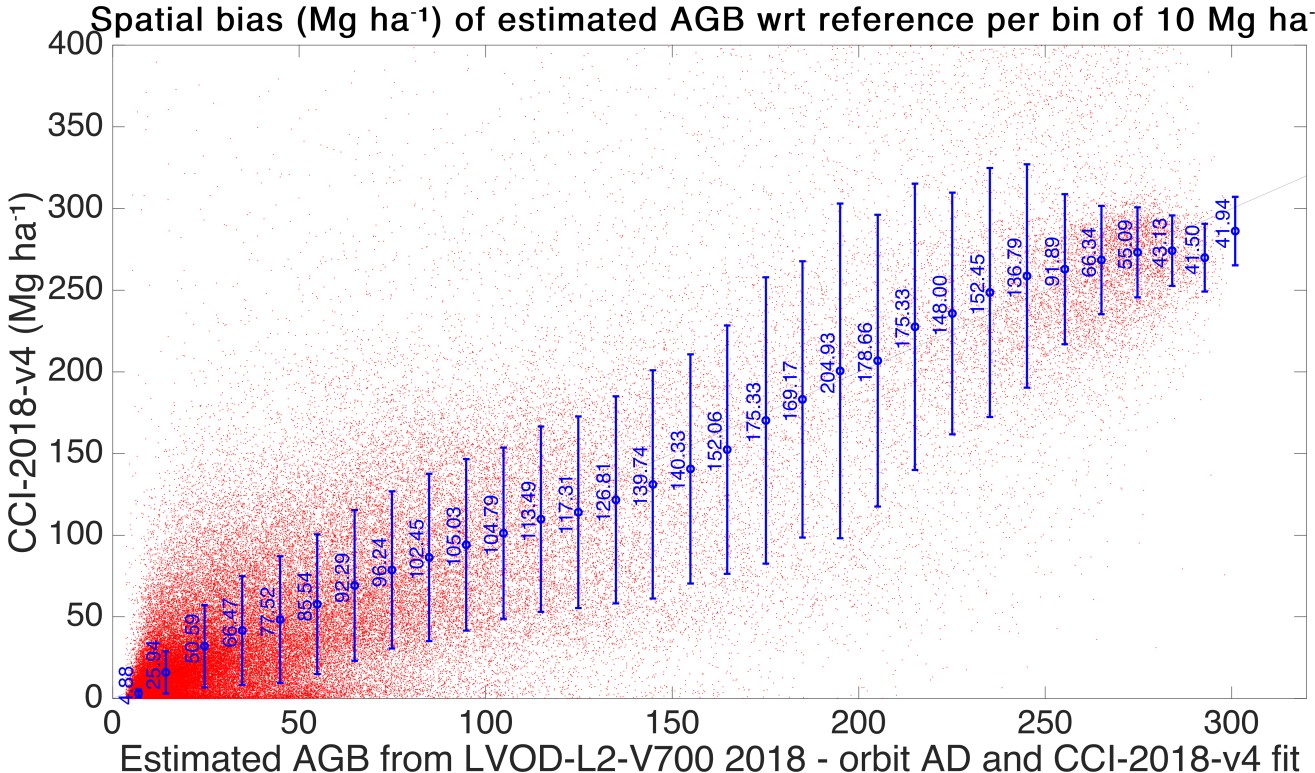

**Figure 4.** Computation of the *std* (blue vertical bars) of the differences between estimated AGB and reference AGB in 10 Mg ha$^{-1}$ bins (blue dots).

# 4 Results

This section presents the analysis of steps 3 and 4 in Fig. 3. In particular, we evaluate the impact of our methods in terms of spatial and temporal scales. The estimated AGB are then evaluated with existing dataset to assess the uncertainty of the produced dataset.

## 4.1 AGB - VOD calibration

Previous to any AGB estimation, the calibration function between the L-VOD and the AGB needs to be estimated. This
calibration function (step 3 of Fig. 3) is presented in Fig. 5. The left panel displays the logistic function calibration between the 2018 L-VOD average and the CCI 2018 for all pixels, mixing ascending and descending orbits. The right panel maps the AGB estimated from the 2018 averaged L-VOD and the calibrated logistic function. The estimated AGB reaches a maximum of approximately 300 Mg ha$^{-1}$ over the tropical forests (Amazonia, Congo, Philippines). Boreal forests, in the northern high latitudes, present AGB estimates around 100-150 Mg ha$^{-1}$. Temperate and arid regions show an estimated AGB lower than 50
Mg ha$^{-1}$. The spatial distribution is coherent with CCI 2018 (see Fig. 2) and with the input L-VOD (see Fig. 1). The Sahara




desert does not present any AGB estimate as no L-VOD retrieval is performed over this region for the L2 product. The Middle east and central Asia are masked out because of high RFI contamination and null AGB calibration data. The white areas in central Australia, south western Africa, and south western America are caused by the filtering of the AGB reference values set to 0.

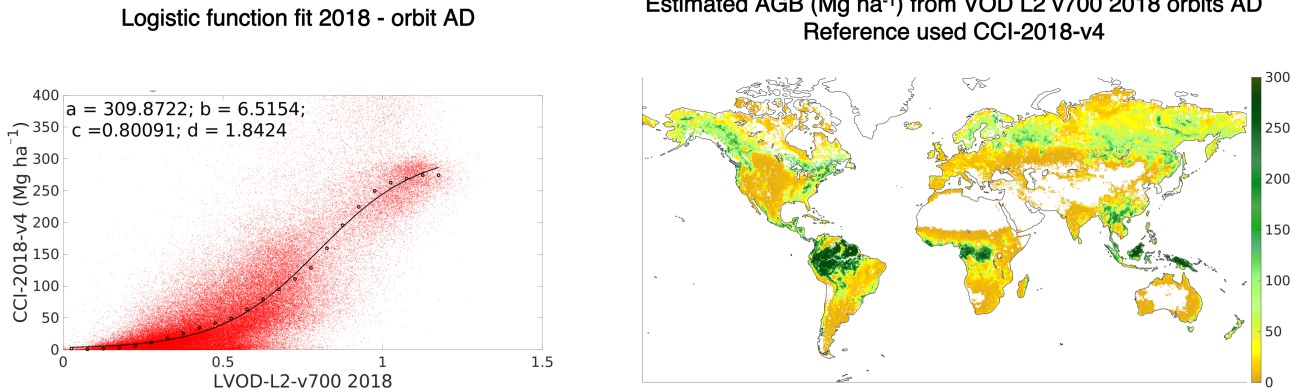

**Figure 5.** The left panel shows the parameter values and the mean logistic fit between the CCI 2018 AGB map version 4 and the VOD from SMOS L2 products filtered and averaged over 2018. The right panel is the estimated AGB in Mg ha$^{-1}$ by applying the logistic function with the estimated $a$, $b$, $c$ and $d$ parameters to the 2018 averaged VOD

The following sections detail the comparison of similar regressions conducted under the different parametrizations described in Sect. 3.

## 4.2    Influence of ascending and descending orbit local time

Considering ascending and descending orbits separately or altogether gives very close calibrations of the parameters in Eq. 1, as reported in Fig. 6. Most importantly, the estimated AGB with three orbital aggregations have similar performances in terms 245   of Bias, R and RMSE compared to the CCI 2018 (Table 1).



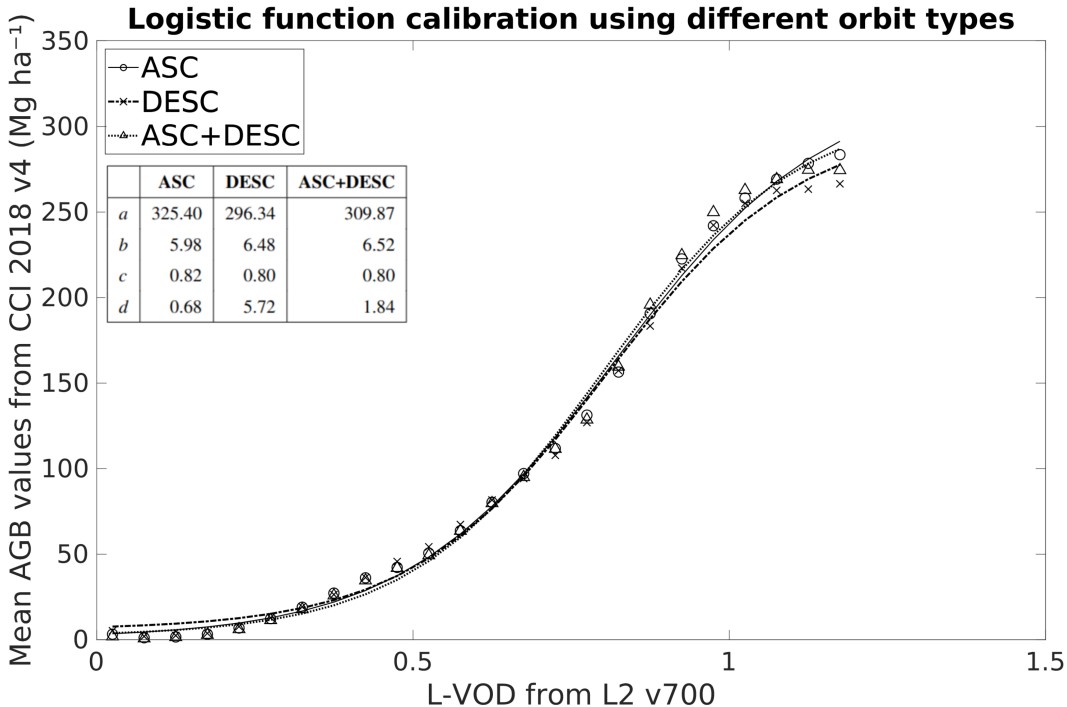

**Figure 6.** Logistic function fit calibrated with AGB from CCI 2018 (Mg ha$^{-1}$) and L-VOD averaged over 2018 using ascending (ASC) and descending (DESC) orbits separately or altogether. The table gathers the parameter values for the three logistic fits.

**Table 1.** R, bias (Mg ha$^{-1}$) and RMSE (Mg ha$^{-1}$) of the AGB estimated with the VOD from different orbit types with respect to CCI 2018

| Orbit | ASC | DESC | ASC+DESC |
|---|---|---|---|
| **R** | 0.85 | 0.84 | 0.86 |
| **bias** | -0.24 | 0.10 | -0.64 |
| **RMSE** | 42.71 | 44.83 | 40.88 |

The correlation coefficients between the estimated AGB and the reference values from CCI 2018 are greater than 0.84 and are slightly higher when combining ascending and descending orbits altogether. Merging both orbit types also marginally decreases the RMSE which is around 40-45 Mg ha$^{-1}$ in all configurations. The bias does not improve when combining the orbit types but remains around the same order of magnitude. This bias is below 1 Mg ha$^{-1}$ in absolute value. Globally, the
performances are not impacted by the orbit type. Considering these results, ascending and descending orbits are mixed to compute the yearly L-VOD maps. It increases the number of daily VOD to compute the average and fills areas where one orbit or the other is affected by RFI.



### 4.3 Impact of a regional calibration

One important feature of this study is assessing the impact of using a single global relationship rather than several regional ones.
Four regions (North, Tropics, Amazon, and Africa) are considered in the analysis. They are displayed as colored rectangles on Fig. 1. The AGB estimates obtained from the calibrated global logistic function (left panel of Fig. 5) are evaluated against the AGB estimates computed from the regionally calibrated logistic functions.

The statistics in terms of R, bias and RMSE obtained with the regional and global calibrations for the four regions with respect to CCI 2018 are displayed on Fig. 7. The regional calibration does not impact R. The bias and RMSE are always lower with the regional fits than the global fit. The improvement is particularly important for the African continent compared to the three other regions. For Africa, the regional bias is close to zero whereas the global bias is equal to -12.61 Mg ha$^{-1}$. The regional RMSE is lower than the global one by 14 Mg ha$^{-1}$. Figure 7 also shows that the difference distribution in Africa between the estimation and CCI 2018 is more centered around 0 when using the regional calibration than with the global one. Also, the bin representing the differences higher than 150 Mg ha$^{-1}$ is not present with a regional calibration for Africa.

For the three other regions, the regional and global RMSE and biases are very similar, the difference in RMSE being less than 1 Mg ha$^{-1}$. A regional relationship is not justified for these regions.

An additional test was conducted for the particular case of the northern high latitudes (above 60° N). The boreal forests are prone to strong seasonality with an extensive snow cover in winter. The impact of snow covered acquisitions on the averaged VOD and the calibration was then evaluated. To this end, the VOD averaged over July and August was compared to the full year average. The July-August averaged VOD globally presents higher values than the full year average. The maximum of the absolute VOD differences is 0.5 but 75 % of these differences lie under 0.042. Ultimately, these differences do not significantly impact the logistic function calibration and the estimated AGB values: 75 % of the differences lie under 8 $Mg ha^{-1}$) which is well below the spatial bias (see Sect. 4.5). It was then chosen to keep winter VOD in the yearly average to take advantage of the full range of observations.





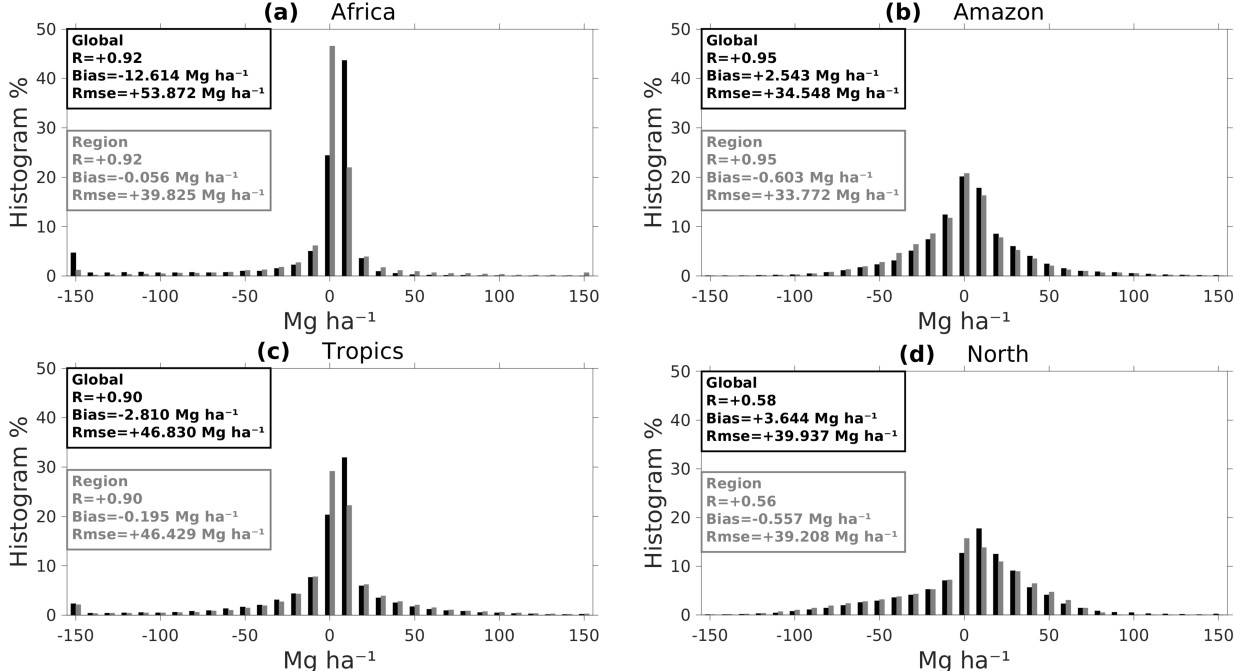

**Figure 7.** AGB differences between the estimation and the calibration data from CCI 2018 in the 4 regions identified in Fig. 1. On each panel, the black histogram represents the distribution of the differences in the study region when estimating the AGB from the global fit (Global R, Bias and Rmse black box) and the grey histogram represents the distribution of the differences when estimating the AGB from the regional calibration (Region R, Bias and Rmse grey box)

## 4.4 Temporal analysis

### 4.4.1 Comparison of the estimated AGB with CCI AGB

To assess the relevance of extrapolating a spatial relationship over time, the VOD-derived AGB with a global calibration is compared with the CCI AGB available for 2017 to 2020. As shown in Table 2, the global statistics characterizing the differences between the estimated AGB and the CCI AGB values are uniform over the years. Spatially, the distribution of the differences are similar across the four years as shown by Panels (a) to (d) in Fig. 8. The region with the highest differences (more than 500 pixels with differences greater than 200 Mg ha$^{-1}$) is the equatorial part of Africa.

Additionally, the time series of the total AGB sum from both datasets over the global area and the African continent are compared against each other on panels (e) and (f) of Fig. 8. The total AGB sum corresponds to the AGB summed over all pixels contained in a given region multiplied by the area of a pixel (always 625 km$^2$ for EASE2 at 25 km). Only pixels common to both datasets and all 4 years are taken into account. Comparing the sum of both AGB estimates is particularly relevant, as the total AGB in a given area is often used as a proxy to the live vegetation biomass carbon stock (Liu et al., 2015; Xu et al., 2021a).





Overall, the CCI reference values are steady over time. On the global scale, the SMOS-derived AGB shows similar time series, presents more variability, and slightly lower estimations compared to the CCI. This agrees with the negative biases from Table 2. For the African continent, the AGB estimated from the VOD is on average 19Pg lower than the reference because of the spatial differences over the equatorial part of the region.

**Table 2.** R, bias (Mg ha$^{-1}$) and RMSE (Mg ha$^{-1}$) between the estimated AGB and AGB values from CCI for the years 2017-2020. AGB was estimated using the global logistic fit computed from the VOD and the CCI AGB for the year 2018. *: year of reference.

|       | 2017  | 2018* | 2019  | 2020  |
|-------|-------|-------|-------|-------|
| R     | 0.88  | 0.86  | 0.87  | 0.87  |
| bias  | -0.62 | -0.64 | -0.24 | -0.65 |
| RMSE  | 38.47 | 40.88 | 39.37 | 40.04 |





**Figure 8.** (a)-(d) Difference in Mg ha$^{-1}$ between the yearly AGB estimated from the 2018 global logistic fit (Fig. 5) minus CCI AGB values for the years 2017-2020. (e)-(f) For the Global and Africa regions defined in Fig. 1, time series of the sum of the AGB from CCI (solid line) and the estimation using LVOD and CCI-2018 (dashed line).

### 4.4.2 Comparison of the estimated AGB with Xu AGB

Besides the SMOS AGB estimates from this product, Xu et al. (2021a) is the only dataset covering more than ten years (from 2000 to 2019). A comparison of the yearly datasets over a 9-year time series (2011-2019) is performed.





The Xu estimates resampled onto the EASE 2 grid at 25 km, are summed according to the method presented in Sect. 4.4.1 and then the sum is divided by 0.49 to convert from Mg of carbon (MgC) to Mg (Xu et al., 2021a).

The comparison of the time series between Xu et al. (2021a) and the estimated AGB using the global fit is displayed in Fig. 9 for all regions of interest. For all cases, the AGB estimates from SMOS VOD, tend to be lower than Xu et al. (2021a) with a higher -though not significant- inter-annual variability. Over the Amazonian rainforest, AGB estimates from SMOS VOD are

on average 25Pg lower than Xu et al. (2021a). This important offset is caused by the differences between Xu et al. (2021a) and the CCI AGB. The latter, being used to calibrate the VOD is systematically lower than the former in forested areas, both in tropical and boreal regions (see panel (f) of Fig. 2).

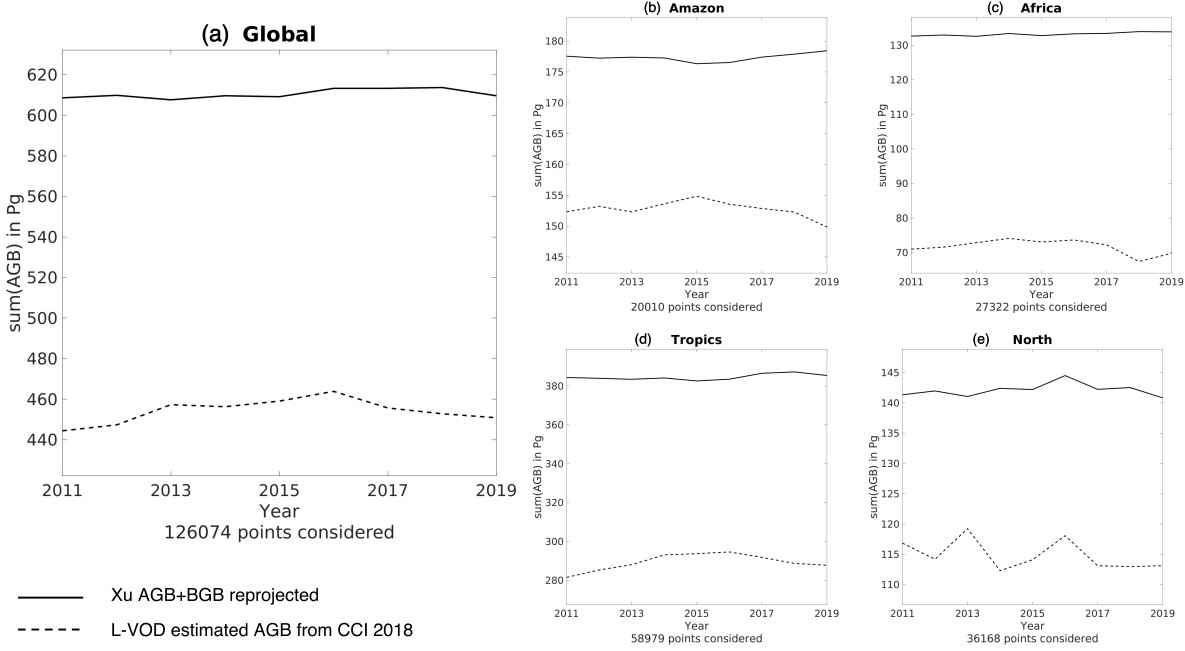

**Figure 9.** Time series of the total sum of the AGB for the different regions defined in Fig. 1 from Xu et al. (2021a) (solid line) and the estimation using SMOS and CCI-2018 (dashed line)

## 4.5    Standard Deviation Computation

The final step of the methodology is the quantification of the uncertainty of the estimated AGB. As shown by Fig. 10 and

Fig. 11, the uncertainties inherited from the input data are negligible compared to the spatial bias component dominating the logistic fit errors. The propagation of the reference AGB standard deviation (blue shaded area on Fig. 11) ranges from $0$ to $5$ Mg ha$^{-1}$. Indeed, the AGB value may vary significantly per pixel from one Gaussian draw to the other but on average, the mean AGB/VOD distribution remains the same considering the number of points taken into account. The same result prevails for the propagation of the mean $VOD_{DQX}$ (red shaded area on Fig. 11). The spatial bias is much more dynamic and ranges from $0$



to 100 Mg ha$^{-1}$ (Fig. 10). As the standard deviation of the input AGB map is not always available and does not impact the
AGB estimation, only the spatial bias component of the logistic fit (Fig. 10) is written to the output product. This component
corresponds to half of the difference between the quantiles 84 and 16 and should be interpreted as a confidence interval around
the mean estimated AGB value provided in the product.

### Spatial Bias (Mg ha⁻¹) of the estimated AGB wrt the reference AGB
### Reference used CCI-2018-v4

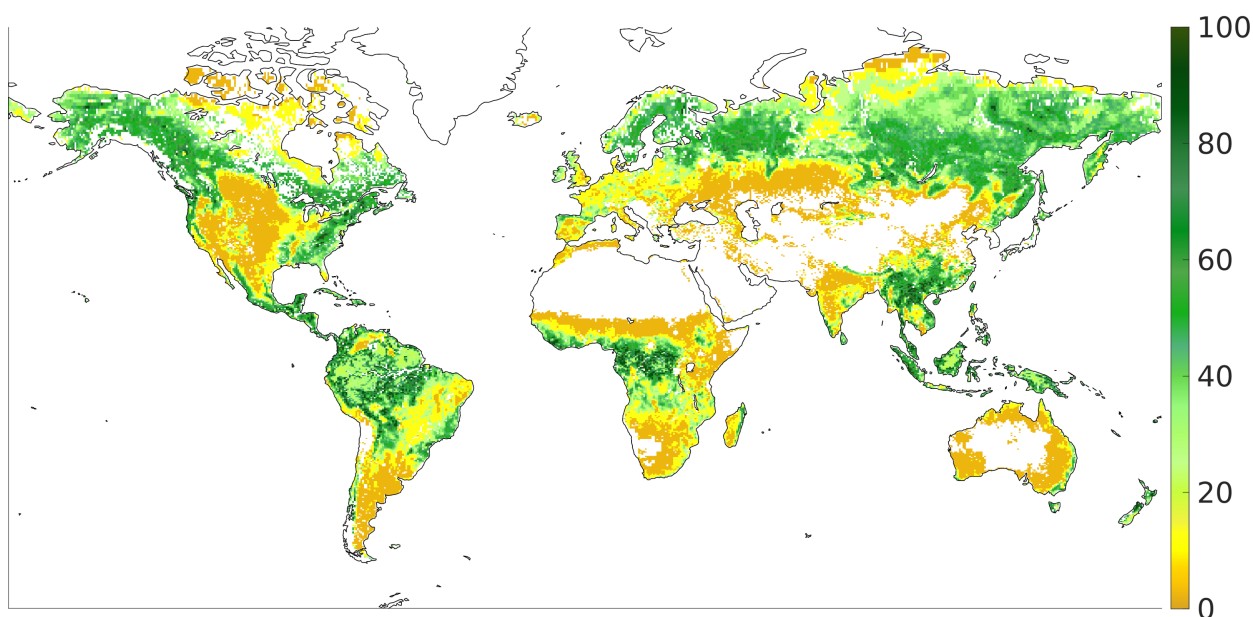

**Figure 10.** Spatial bias (Mg ha$^{-1}$) related to the logistic fit between the AGB values from CCI 2018 and the SMOS L-VOD.

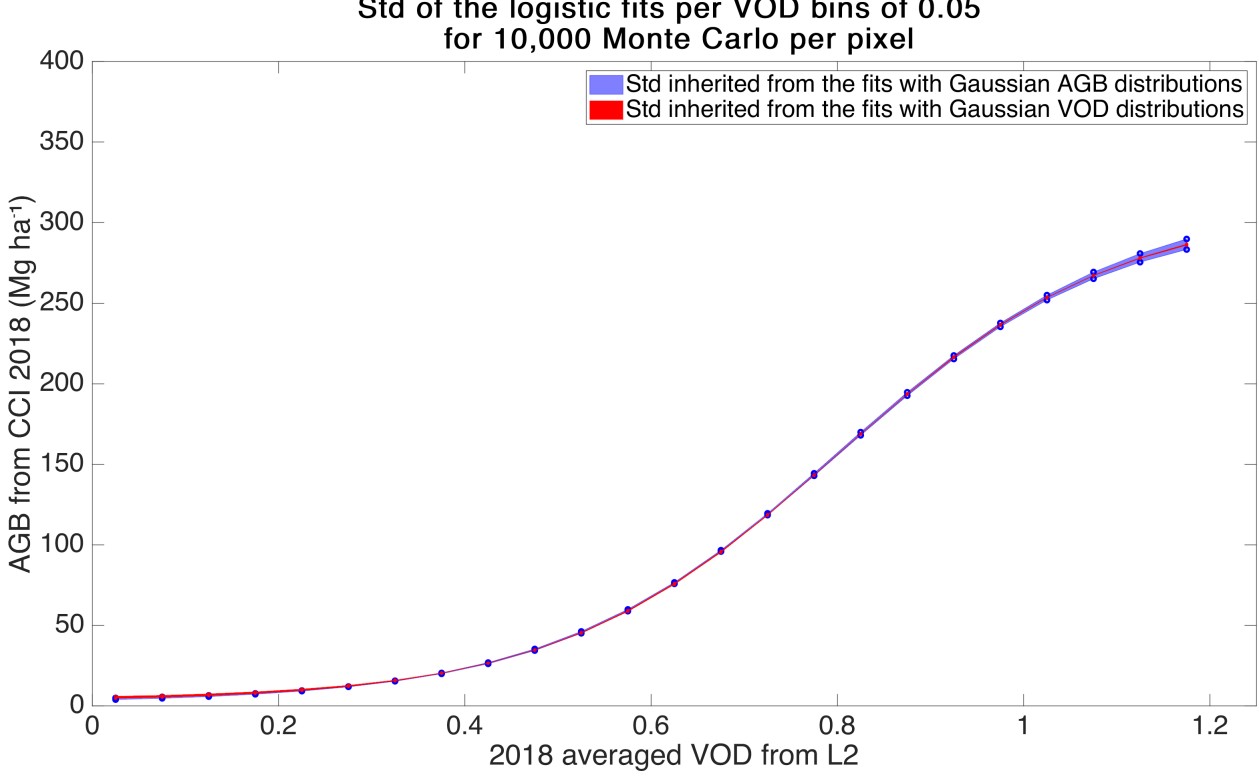

**Figure 11.** Standard deviation (Mg ha$^{-1}$) of the 10,000 Monte-Carlo draws for a Gaussian AGB distribution per pixel (blue) and a Gaussian L-VOD distribution per pixel (red).

## 5    Discussion

The paper describes a dataset which is a follow-up to the work initiated by Liu et al. (2015) at high frequencies (X/C band) and developed at L-band for SMOS by Rodríguez-Fernández et al. (2018); Fan et al. (2019); Mialon et al. (2020). This analysis goes further and quantifies the effects of different aspects of the method on the derived AGB which are i) mixing the morning/afternoon overpasses, ii) using a relationship at the global scale, iii) extrapolating a spatial relationship over time, and most importantly iv) providing a confidence interval range of the estimated AGB. First, the consistency of the approach is confirmed

by comparing the L2 product performances with two other SMOS-derived VOD: the CATDS L3 product (Al Bitar et al., 2017) and the INRA-CESBIO (IC) dataset (Fernandez-Moran et al., 2017). This part is not detailed in the present analysis as the results are very similar. Even though the three SMOS VOD are derived with different algorithms ((Kerr et al., 2012; Al Bitar et al., 2017; Fernandez-Moran et al., 2017)), the approach developed in this paper leads to equivalent performances (as shown in Table 3). The IC algorithm shows higher R (0.89 vs 0.87 for L3 and 0.86 for L2) and slightly lower RMSE and bias. Nev-

ertheless, taking into account the order of magnitude of the bias and RMSE differences across the 3 products (0.9 Mg ha$^{-1}$





in bias and 4 Mg ha$^{-1}$ in RMSE), they are completely negligible in comparison to the order of magnitude of the spatial bias introduced by the fitting of the logistic function (Fig. 4, 30-90 Mg ha$^{-1}$).

**Table 3.** R, Bias (Mg ha$^{-1}$) and RMSE (Mg ha$^{-1}$) between the AGB estimated with the VOD from L2, L3 and IC mixing ascending and descending orbits and the CCI 2018 calibration data

| Product | L2 v700 | L3 v339 | IC v105 |
|---------|---------|---------|---------|
| R | 0.86 | 0.87 | 0.89 |
| bias | -0.64 | -1.33 | -0.39 |
| RMSE | 40.88 | 39.14 | 36.43 |

The effect of the time of observation is also evaluated. It is admitted that morning overpasses (6 am for SMOS) offer more stable surface conditions as the Earth's surface reaches a thermal equilibrium. Therefore, better (SM, VOD) retrievals are expected using the morning orbits. It has however no impact for the present application as the three cases (i.e. only morning, only afternoon, and the two combined) have similar performances, with a correlation coefficient R ranging between 0.84 and 0.86 (see Table 1). This results from the yearly averaging that smooths the daily variability, caused by the vegetation water content. Both overpasses (morning and afternoon) are then merged, as it increases the number of observations per pixel to compute the yearly VOD average.

Then, the impact of using a global relationship at the regional scale is estimated. Across the four studied regions (Fig. 1), only Africa presents a significant bias and RMSE improvement when using a regional relationship (Fig. 7). The case of the African continent is specific, as the equatorial part of the region has the highest CCI AGB values, up to 400 Mg ha$^{-1}$. Deriving a regional logistic relationship between the VOD and the AGB reproduces these high values better. With the global calibration, the logistic fit does not reach the 400 Mg ha$^{-1}$ threshold (see left panel of Fig.5) and saturates around 300 Mg ha$^{-1}$. This value corresponds to the average AGB value in the Amazon rain-forest according to the CCI 2018 map. Hence, applying the global relationship over Africa leads to lower estimated AGB than the CCI AGB. Moreover, the VOD values are more scattered and present more variability in Central Africa compared to the VOD values in the Amazon rain-forest. However, these derived AGB estimates in the equatorial part of Africa are consistent with other AGB datasets such as Avitabile et al. (2016). Adopting a single global logistic relationship between the SMOS LVOD and the input AGB is a good trade-off between performance, simplicity and consistency.

Regarding the time dimension, the general method relies on the hypothesis that a spatial relationship for a single year is accurate enough to create the time series. Indeed, the VOD-AGB relationship is defined for a particular year and is propagated over time. This assumption, never evaluated by any studies, is tested thanks to the four years of the CCI dataset. The logistic relationship defined with the CCI 2018, was applied to the SMOS VOD for 2017, 2019, and 2020. The estimated AGB are then compared to the CCI 2017, 2019 and 2020 respectively. The performances of the SMOS-derived AGB are similar for all years in terms of coefficient of correlation (between 0.86/0.88 according to Table 2) and RMSE (between 38 and 41 Mg ha$^{-1}$). The biases are also very close (around -0.6 Mg ha$^{-1}$) with better results for 2019 (-0.24). It supports the consistency



of the VOD-AGB relationship. The main differences between SMOS-derived AGB and the CCI AGB are observed over the tropical forests in Africa, and the boreal forests of eastern Siberia. In Africa, the lower estimates are consistent for all years and

are caused by the global VOD-AGB relationship (see previous comment). In this study, the CCI AGB map for 2018 has been used to optimize the spatial logistic relationship. The dynamic of the CCI AGB over time is low, as shown by the steady black curves on panels (e) and (f) of Fig. 8. Spatially, the AGB differences from one year to the other do not exceed 11 Mg ha$^{-1}$ in absolute value for 99 % of the EASE 2 pixels. These differences fall well below the uncertainties associated to the CCI maps (up to 200 Mg ha$^{-1}$) which already have a low influence on the AGB estimates as demonstrated by the Monte-Carlo test (Sect.

4.5). Consequently, fitting the relationship with the CCI map from another year does not significantly impact the estimation of the AGB time series.

To support the analysis, the derived time series is then compared to Xu et al. (2021a), which is the only other dataset covering ten years. Differences between both datasets are expected. Indeed, Xu et al. (2021a) method includes lidar/optical/radar dataset whereas SMOS is a passive microwave sensor. Moreover, Xu et al. (2021a) estimate the Living Biomass (dry above and below

ground) whereas SMOS-derived dataset corresponds only to the dry above-ground biomass. This explains the higher values observed for Xu et al. (2021a) (Fig. 9). Such differences are aslo observed between Xu et al. (2021a) and the CCI. The time series show more inter-annual variability for the SMOS derived dataset. This variability is low, ~4 % (difference of 18 Pg over an average value of ~450 Pg) at the global scale (left Fig. 9, dotted line). Differences exist between the trends. For example, there is an opposite trend in the Amazon region or a peak in the northern region in 2013, not observed in Xu et al. (2021a)

dataset. Such differences are expected due to the above-mentioned reasons, and further investigations are needed to explain these divergences.

Finally, this new dataset provides a confidence interval on the derived AGB, contrary to other similar databases. The restitution error is mostly dominated by spatial biases which highlights the inability of a single global model to map all the relationships between the VOD and the AGB. Fig. 4 presents the estimated confidence interval per bin of SMOS-derived AGB.

Fig. 10 is the same information projected on a map to better visualize the spatial distribution of this bias. The propagation of the uncertainties of the input data through the logistic function optimisation was also carefully checked using a Monte-Carlo approach. The impacts of the VOD and CCI AGB errors on the AGB estimation were evaluated and found to be small (5 Mg ha$^{-1}$ for the highest AGB values) compared to the AGB error. Fig. 4 also emphasizes that strong AGB values may be underestimated when computing AGB from the L-VOD and the optimized logistic function. Indeed, L-VOD tends to saturate

over densely vegetated areas, even though it does not saturate as much as optical indices or VOD in C and X bands.

## 6   Conclusions

The paper presents the AGB dataset estimated from the SMOS L-band passive microwave VOD, which is directly related to the vegetation water content (Konings et al., 2019). When averaged over a year, it nevertheless constitutes a good proxy to the AGB (Rodríguez-Fernández et al., 2018), and presents the advantage of covering a long-time series, starting in 2011.

This study focuses on the method and analyses the robustness of the approach to estimate the AGB from the SMOS level 2

VOD. In particular, it is shown that the AGB estimates do not depend on the over-passing time of the satellite. Moreover, even though a global relationship (VOD-AGB) is appropriate, slight differences are observed over Africa compared to using a dedicated relationship for this continent. Indeed, it is shown that the global VOD-AGB calibration underestimates the high AGB in the tropical region of Africa. The analysis also evaluates the multi-year dataset with two global time series. The SMOS-derived

AGB dataset is very close to the CCI AGB used for its calibration. The estimated AGB are significantly lower than the one from Xu et al. (2021a) and presents more inter-annual variability. The former is due to Xu et al. (2021a) measuring the living biomass whereas the presented method leads to an estimate of the AGB. The latter is directly inherited from the inter-annual variability of the yearly averaged SMOS L-VOD and from the fact that the two data do not measure the same quantity. Finally, this study provides for the first time an estimation of the error of the derived VOD in order to better assess its application in the

context of biomass monitoring. The present dataset is freely accessible from the CATDS website.

*Data availability.* Data set DOI:10.12770/95f76ff0-5d89-430d-80db-95fbdd77f543. (Boitard et al., 2024)

The AGB estimates from SMOS L-VOD are open access and available at: https://data.catds.fr/cecsm/Land_products/L4_Above_Ground_Biomass/

Data content description:

(Boitard et al., 2023)

*Author contributions.* Simon Boitard performed the investigation, the software development and the figures and tables creation. Arnaud Mialon was in charge of the conceptualization, the supervision and the validation of the study. The methodology was a common work between Simon Boitard, Arnaud Mialon, Nemesio Rodriguez-Fernandez and Stéphane Mermoz. Simon Boitard and Philippe Richaume

conducted the formal analysis of the study. The input, output and some of the auxiliary data curation was jointly done by Simon Boitard, Arnaud Mialon, Nemesio Rodriguez-Fernandez, Philippe Richaume and Stéphane Tarot. The original draft was written by Simon Boitard and Arnaud Mialon and all other co-authors except Stéphane Tarot contributed to the review and editing of the manuscript. Yann H. Kerr and Arnaud Mialon administrated the project and Yann H. Kerr acquired the funding necessary to this study.

*Competing interests.* The authors declare that they have no conflict of interest.

*Acknowledgements.* The authors thank the CNES (Centre National d'Etudes Spatiales), the CATDS (Centre Aval de Traitements des Données SMOS) supported by CNES and IFREMER, CNES TOSCA (Terre Océan Surface Continentale Atmosphère) program, and ESA for the SMOS mission and for funding the OSMOSE project (PM-VOS PM-VOS, ESA AO/1-10908/21/NL/IA)



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
