# Peer review of "Above ground biomass dataset from SMOS L band vegetation optical depth and reference maps"

_Earth System Science Data, 2024_

## Community Comment (CC1)

Net change during 2011-2019

Net change (MgC ha$^{-1}$ yr$^{-1}$)

---

## Author Response (AR1)

**Responses to Anonymous Referee #1**

We thank the referee #1 for reviewing and commenting on the manuscript. This valuable feedback was carefully taken into account to improve the quality of the paper.

Below, Referee #1 comments are marked in red.
Responses to the comments are marked in blue.
Changes that have been made in the manuscript are marked in *italic*.

Major comments:

1. A definition of AGB needs to be added to the introduction. Does it refer solely to forest above-ground biomass? Furthermore, significant work regarding the derivation of AGB from microwave remote sensing needs to be included, e.g., using brightness temperature. Additionally, specific properties of L-VOD are unfairly attributed only to SMOS L-VOD, while significant literature has been put out on using SMAP L-VOD for similar applications.

   Indeed, from the introduction, it was not clear whether AGB referred to the forest or the entire vegetation Above Ground Biomass. We added a precision in the introduction, line 29.
   Modifications:
   *lines 29-30: The total Above Ground Biomass (AGB, which accounts for the entire land vegetation biomass), is strongly linked to the carbon content in the vegetation (Losi et al., 2003; Djomo et al., 2011).*

   Some references were indeed missing and we focused too much on the SMOS L-VOD in the introduction, whereas this part is supposed to give a global state of the art. We mentioned and added SMAP references in the introduction.
   Modifications:
   *line 53-54: This radiative parameter is also derived at L band from the Soil Moisture Active Passive (SMAP) mission (Entekhabi et al., 2010; Konings et al., 2017; Chaubell et al., 2021)*
   *+ Deletion of all references to SMOS L-VOD in the 4th paragraph of the introduction*

   We also added references on using brightness temperatures from the SMOS and SMAP mission to derive AGB.
   Modifications:
   *lines 65-67: [...] used several static AGB maps to derive spatial relationships between VOD datasets and reference AGB estimates while Prigent and Jimenez (2021); Salazar-Neira et al. (2023) directly established a link between these static AGB maps and brightness temperatures using neural networks.*

2. The authors need to explain how the reference maps were aggregated to the SMOS spatial resolution. The aggregation method might be important, particularly in the case of CCI AGB, which comes in a resolution of 100m x 100 m. How were the uncertainties (standard deviations) used in the context of aggregating to the SMOS resolution?

   The data section lacked precisions about the aggregation method. A new sentence has been added at the beginning of part 2.2.
   Modifications: *lines 124-125: The AGB reference maps were regridded from their native sampling to the same EASE grid version 2 25 km as the interpolated L2 products. The regridding method was a*

*weighted average of all non-nodata contributing pixels.*

The uncertainties associated with the CCI AGB maps were not used to aggregate to the SMOS resolution. However, the CCI uncertainty (standard deviation) maps were also aggregated to the SMOS resolution by a quadratic mean of all non-nodata contributing pixels. This aggregated uncertainty map was in turn used to quantify the impact of the reference AGB uncertainty on our AGB estimates. As shown on Fig. 11 and Fig. 10, the impact is small (0-5 t/ha) compared to the spatial bias inherited from the logistic fit (0-100 t/ha).
Modifications:
*lines 225-226: The CCI uncertainty (standard deviation) maps were aggregated to the SMOS resolution by a quadratic mean of all non-nodata contributing pixels.*

3. The methods chapter is not very clear.
   ○ The authors mention that all years of SMOS data and other AGB reference maps were used for the final dataset production. However, the paper describes only the methodology applied for 2018, a year where both CCI AGB and SMOS L-VOD estimates are available. How did the authors proceed with the years with no available reference maps?

   The general explanations on the method were not clear. The problematic sentence (line 162) was reworded. In a general way, we calibrate the LVOD-AGB spatial relationship for a reference year, where, indeed, both an AGB reference map and SMOS L-VOD estimates are available. Then, we extrapolate this spatial relationship over time to derive a time series. We hope that after the below modifications it is more clear.
   Modification:
    lines 169-171: *Once the spatial calibration of the AGB-LVOD relationship is thoroughly studied for the reference year, it is extrapolated over time to all other SMOS years. The same procedure is repeated to create AGB time series estimates from other AGB reference maps.*
   *Update of Fig. 3:*
   *3- Fitting (for a reference year)*
   *4- Validation: Addition of 'Temporal extrapolation of the spatial calibration'*

   ○ Were all CCI AGB maps used? CCI AGB provides change maps with a quality flag concerning the reliability of AGB change. Where these flags used?

   To produce the AGB estimates from the CCI, only the year 2018 was used to calibrate the spatial relationship. We did not need the CCI quality flags to calibrate the spatial relationship. However, CCI 2017 to 2020 maps were used for the validation. In particular, in section 4.4.1 we compared our estimates using CCI 2018 as a reference with other CCI years and found the global statistics characterizing the differences between the estimated AGB and the CCI AGB values to be uniform over the years (Table 2).
   Following the referee's suggestion, we have downloaded all 10°*10° quality flag tiles for the three provided change maps (2020-2019, 2019-2018, 2018-2017) that are relevant to the evaluation shown on Fig. 8. Then we aggregated the quality change maps to the SMOS resolution. To aggregate the flags to the SMOS resolution, we selected the value which appears most often of all the sampled points within à 25*25km² pixel. After such an aggregation, it appears that the vast majority of pixels in the change map is set to "Improbable change" as shown below:
   Aggregated Quality flag at 25km for the 2018 - 2017 change map:

[Figure]

Aggregated Quality flag at 25km for the 2019 - 2018 change map:

[Figure]

Aggregated Quality flag at 25km for the 2020 - 2019 change map:

[Figure]

We also checked that the aggregation was appropriately done by checking some native resolution tiles and it appears that most of the pixels are indeed set to 'Improbable change':

CCI change tile

N00W060_ESACCI-BIOMASS-L4-AGB-MERGED-DIFF_QF-100m-2020-2019-fv4.0 2020-2019

over Brazil:

[Figure]

CCI change tile N70E020_ESACCI-BIOMASS-L4-AGB-MERGED-DIFF_QF-100m-2020-2019-fv4.0
2020-2019 over Finland:

[Figure]

Following these results, we concluded that unfortunately the quality flags could not be practically used for our study at 25km.

○ How was the Avitabile map used? Is there any bias between CCI AGB and Avitabile AGB? Furthermore, Avitabile corresponds to 2010, while Fig. 3 shows yearly AGB estimates starting in 2011. For what year was Avitabile used?

This point was also not well explained. The Avitabile map is used as another reference map to produce another estimated AGB dataset (NetCDF file, independent from the CCI estimates), also available from the CATDS website. Our principle is to create different datasets from different AGB reference maps (see a comment below concerning validation). For Avitabile, we calibrated the logistic fit with the SMOS average from 2011, as 2010 is too polluted with RFI

and not complete (as the First 5 months of 2010, were dedicated to the commissioning phase). The bias between CCI2018 and Avitabile reference maps is displayed on Fig. 2d. Modifications: lines 171 - 175: *For now, the only other AGB reference map used in this study is the one from Avitabile et al. (2016), representative of the year 2010. This map is hence linked to SMOS L-VOD from 2011 as 2010 does not meet the quality requirements (see Sect. 2). Ultimately, there are as many AGB time series estimates as reference maps used. Currently, there is one NetCDF file holding the AGB time series estimates from the CCI reference map and one NetCDF file holding the AGB time series estimates from the Avitabile et al. (2016) reference map.*

4. The evaluation is very minimal, particularly in the temporal domain. Given that the study provides a dynamic AGB dataset, the plausibility of temporal patterns is paramount. The authors should consider validating change patterns, e.g., per land cover.

Indeed, validation of AGB maps and time series is a tedious task. So far, except the attempt of Araza et al. (2023), no real evaluation protocol of AGB maps exists. As for the VOD-derived estimates of AGB, the temporal trends have never been properly evaluated. Even though optical indices have been used to monitor the vegetation health and status over long time series, the validation of the L-VOD and derived AGB cannot be properly based on them. Many studies showed that the optical indices saturate over dense forests, whereas the L-band penetrates more into the vegetation layer. Moreover, optical observations are sensitive to the leaves (greenness or browning) whereas the L band is more sensitive to trunk and branches. For instance Bousquet et al. (2022) compared L-VOD with the Enhanced Vegetation Index (EVI) and showed that particular events, like fires, led to different temporal signatures depending on the vegetation class. Another validation option found in some studies is to link the AGB variations to external events affecting the vegetation biomass in forests or other specific biomes. For example, Wigneron et al (2020) explained the variability of AGB in 2015/16 in Amazonia with the El Nino events. This is a possible explanation but does not constitute rigorous validation. This is why releasing different AGB datasets produced from different L-VOD data and using different methods is fundamental. Comparing the datasets against each other offers the community an alternative way to validate the trends.

Nevertheless, following the reviewer comment, we attempted validating our estimates for two specific regions with different biomes.
First, Berner et al. (2020) studied the greening of the Arctic tundra biome from 1985 to 2016 with the NDVI from Landsat and provide a time series of the NDVI anomaly over these years (see below figure directly extracted from their article)

[Figure]

**Fig. 2 Tundra greenness and summer air temperature time series and covariation.** Left panels show changes in mean Landsat NDVI$_{max}$ [unitless] anomalies for the Arctic and each zone from 1985 to 2016 (**a**) and 2000 to 2016 (**d**). Middle panels show changes in mean summer warmth index [SWI; °C] anomalies from 1985 to 2016 (**b**) and 2000 to 2016 (**e**) derived from five temperature data sets. Right panels show the relationship between mean Arctic NDVI$_{max}$ and SWI anomalies from 1985 to 2016 (**c**) and 2000 to 2016 (**f**). Spearman's correlation coefficients ($r_s$) relating NDVI$_{max}$ and SWI are provided for each period. Narrow error bands and whiskers depict 95% confidence intervals derived from Monte Carlo simulations ($n = 10^3$). Note that while mean SWI time series are based on pan-Arctic data, the NDVI$_{max}$ time series, and NDVI$_{max}$–SWI relationships are based on sites where Landsat data were available from 1985 to 2016 (**a**, **c**) and 2000 to 2016 (**d**, **f**), as shown in Fig. 1.

In order to compare with our AGB estimates, we selected the points located in the Arctic from our estimates and Xu et al. maps (converted to AGB, see our reply to the major comment from the second reviewer) thanks to the following mask (Dark blue: 'High Arctic', Light blue: 'Low Arctic', Yellow: 'SubArctic'):

[Figure]

We then plotted the anomaly of the total AGB sum time series within the Arctic for both datasets (our AGB estimates and Xu et al) between 2011 and 2016 to compare it with the NDVI anomaly over the same time period:

[Figure]

Both AGB time series do not correspond well with the NDVI time series between 2011 and 2016, which is a well-known fact (Grant et a (2012)l)

We also conducted the same analysis over the equatorial part of Africa to compare with Nooni et al (2024). They studied the NDVI time series between 1982 and 2021 in the area located within [2 20]°N and [18° W - 55° E]:

*The inter-annual variations in (**a**) the mean NDVI, (**b**) P, and (**c**) TMEAN at annual and seasonal timescales from 1982 to 2021. The linear trend calculated using the linear regression (dashed lines) trend is calculated using the least squares linear trend fitting method over the period (at p-value < 0.05). Annual (blue color), spring (MAM, orange color), summer (JJA, yellow), autumn (SON, purple color), and winter (DJF, green color).*

[Figure]

**(a)**

Below are the time series anomalies of the AGB sum in Equatorial Africa from our estimates and Xu et al estimates for 2011-2019:

[Figure]

Again, both time series do not show great similarities with the NDVI.

We hope these further analyses help to demonstrate the difficulties of validating the AGB change with optical indices.

Araza et al. 2023, Past decade above-ground biomass change comparisons from four multi-temporal global. International Journal of Applied Earth Observation and Geoinformation, vol. 118, pages 103274, 1569-8432, doi https://doi.org/10.1016/j.jag.2023.103274, https://www.sciencedirect.com/science/article/pii/S1569843223000961

Bousquet et al. 2022, Monitoring post-fire recovery of various vegetation biomes using multi-wavelength satellite remote sensing, Biogeosciences, 19, n 13, pages 3317--3336, doi 10.5194/bg-19-3317-2022 https://bg.copernicus.org/articles/19/3317/2022/

Berner, L.T., Massey, R., Jantz, P. *et al.* Summer warming explains widespread but not uniform greening in the Arctic tundra biome. *Nat Commun* **11**, 4621 (2020). https://doi.org/10.1038/s41467-020-18479-5 https://rdcu.be/dSVFn

Grant, J., Wigneron, J.-P., Drusch, M., Williams, M., Law, B., Novello, N., and Kerr, Y.: Investigating temporal variations in vegetation water content derived from SMOS optical depth, 2012 IEEE International Geoscience and Remote Sensing Symposium, pp. 3331–3334, https://doi.org/10.1109/IGARSS.2012.6350590, 2012.

Nooni, I.K.; Ogou, F.K.; Prempeh, N.A.; Saidou Chaibou, A.A.; Hagan, D.F.T.; Jin, Z.; Lu, J. Analysis of Long-Term Vegetation Trends and Their Climatic Driving Factors in Equatorial Africa. *Forests* **2024**, *15*, 1129. https://doi.org/10.3390/f15071129 https://www.mdpi.com/1999-4907/15/7/1129

Jean-Pierre Wigneron *et al.* Tropical forests did not recover from the strong 2015–2016 El Niño event.*Sci. Adv.***6**, eaay4603 (2020) .DOI:10.1126/sciadv.aay4603 https://www.science.org/doi/full/10.1126/sciadv.aay4603

5. The few time series provided do not agree very well with either CCI AGB (Fig. 8) or Xu AGB (Fig.9). While the authors mention the disagreements in the discussion chapter, further explanations are needed.

Indeed.
Several points can explain the disagreement between our estimations and the CCI and Xu et al.  First of all, these AGB maps were produced from different sensors. As presented in section 2.2.1, the CCI AGB is mainly produced from L and C-band radars (ALOS2 PALSAR2 and Sentinel-1, active instrument) while Xu et al maps are primarily derived from plots, lidar, microwave and optical data (section 2.2.3). Our estimates are only based on the SMOS L-VOD (passive instrument). Radar and passive microwave do saturate over dense vegetation, but at different vegetation density. This point is mentioned in the paper introduction. Radar is also more sensitive to the roughness at the vegetation/air interface. This may also explain the observed bias between Xu et al and CCI (fig 2e).

Moreover, as shown on Fig. 8, the CCI time series are very steady. The CCI biomass project applies a scaling factor to homogenize the yearly AGB estimates, and to limit abrupt changes at the pixel scale from one year to the other (chapter 4.5.3 from the ATBD: https://climate.esa.int/media/documents/D2_2_Algorithm_Theoretical_Basis_Document_ATBD_V4.0_20230317.pdf) . While this approach may be fully justified, it explains some of the temporal differences observed on Fig. 8. Another reason which may explain the variability of our AGB estimates is the natural variability of the L-VOD. Even though we average the L-VOD over a year to minimize the

vegetation water content impact on the estimation, some residuals persist and affect the derivation of the AGB.

Finally, the VOD derived AGB presented in many papers published in Nature Journal (See Fan et al., Nature Geosciences 2019) also presents a time variability different from Xu et al. Contrary to our study, the authors showed the anomalies (one can not evaluate the bias).

See the following figure fom Yang, H., Ciais, P., Frappart, F. *et al.* Global increase in biomass carbon stock dominated by growth of northern young forests over past decade. *Nat. Geosci.* **16**, 886–892 (2023). https://doi.org/10.1038/s41561-023-01274-4. We mention this study as it is suggested in a comment (see the latest comment, further below)

[Figure]

[Figure]

**Fig. 4 | Comparison of the IAV in global carbon fluxes and global atmospheric CGR.** The IAV of reversed global atmospheric CGR (black line), global total carbon fluxes in all forests (solid lines) and young forests (dashed lines) from L-VOD without CWD buffering (red lines) and total carbon fluxes from L-VOD with CWD buffering (blue lines) from 2012 to 2019. The right-hand panel shows the contribution (%) of all forests in five climate biomes to the IAV in global carbon fluxes from L-VOD without CWD buffering (red bars) and with CWD buffering (blue bars). Additionally, the contribution of young forests among all forests is represented by the white bars. The global carbon fluxes (δTB) are calculated by using annual changes in live total biomass compared to the previous year. The shadings of each curve show the uncertainty of CGR and the standard deviation of L-VOD TB estimated by different calibrations.

Modifications:

*lines 373-375: As the CCI biomass team applies a scaling factor to homogenize the yearly AGB estimates and to limit abrupt changes at the pixel scale between years (Santoro et al., 2023), the dynamic of the CCI AGB over time is low (steady black curves on panels (e) and (f) of Fig. 8).*

*lines 380-391: To support the analysis, the derived time series is then compared to Xu et al. (2021a), which is the only other dataset covering more than ten years. Differences between both datasets are expected as they were produced from different inputs and with different methods. Xu et al. (2021a) includes lidar, optical and radar datasets whereas SMOS is a passive microwave sensor. Radar and passive microwaves do saturate over dense vegetation, but at different vegetation densities. Radar is also more sensitive to the roughness at the vegetation/air interface. For every studied region, the SMOS derived time series show more inter-annual variability. This may be partly caused by the natural fluctuations of the yearly averaged L-VOD due to changes in the vegetation water content to which the L-VOD is highly sensitive. Thanks to the yearly averaging, this variability remains relatively low, ~4 % (difference of 18 Pg over an average value of ~450 Pg) at the global scale (left Fig. 9, dotted line). Differences also exist between the trends. For example, there is an opposite trend in the Amazon region or a peak in the northern region in 2013, not observed in Xu et al. (2021a) dataset. Such differences are expected due to the above-mentioned reasons, and further investigations and inter-comparisons between published and future AGB datasets will be needed to explain and understand these divergences*

Minor comments:

Line 34: "satellite remote sensing of AGB" - direct AGB measurements do not exist - change to deriving AGB from satellite remote sensing.

Modifications:
line 36 : *This is why deriving AGB from satellite remote sensing, which provides a global and regular coverage of the Earth surface, complements AGB in situ data very well*

Lines 55-57: The mentioned properties are not inherent only to SMOS L-VOD. Move above to the general properties of L-VOD.
This is linked to major comment 1 and has been dealt with.

Fig. 3: Step 4 "Validation" - Change "different parameters" to, e.g., "different model fits/factors".
Fig 3. has been modified accordingly

Fig. 3: Are the AGB estimates starting in 2011, not 2010?
Yes the time series starts in 2011, for 2 reasons :

- RFI that significantly impacted SMOS in 2010

- the first 5 months (January - May) were dedicated to the commissioning phase, during which tests/calibration were performed, the instrument needed time to stabilize… It is the case for every Earth Observation mission. So 2010 is not complete and not quality checked and hence not included in our dataset.

Modifications:
*lines 90-94: The SMOS Level 2 (L2) products (Kerr et al., 2012) between 2011 and 2023 are used for this study. 2010 is set aside for two reasons. The first 5 months (January - May) were dedicated to the commissioning phase, during which the instrument was stabilizing and extensive tests and calibration were performed. This year was also extremely polluted with Radio-Frequency Interferences (RFI). These RFI are emitted by human-built equipment and mask the Earth natural emission over large areas. They prevent SMOS from globally covering the Earth (Oliva et al., 2016).*

Line 384: 2011, not 2010?
See previous comment.

Abbreviations are not consistent between plots and captions (CCI-2018-v4; CCI AGB reprojected)
Abbreviations have been harmonized between plots and captions in Fig 2., Fig. 3, Fig. 4, Fig. 5, Fig. 8 and Fig. 10.

Fig 10: SMOS L-VOD 2018?
Corrected in Fig. 10 caption

**Responses to Anonymous Referee #2**

We thank the referee #2 for reviewing and commenting on the manuscript. This valuable feedback was carefully taken into account to improve the quality of the paper.

Below, Referee #2 comments are marked in red.
Responses to the comments are marked in blue.

Changes that have been made in the manuscript are marked in *italic*.

This paper describes a dataset of above ground biomass (AGB) obtained from satellite remote sensing. The L-Band Vegetation Optical Depth (VOD) from the Soil Moisture and Ocean Salinity (SMOS) mission linked to existing AGB maps allows defining spatial relationships between L-VOD and AGB. AGB is then determined for 2011-2023 from the L-VOD data. In this study, the impacts of the orbit type (using only ascending or descending data or combining both) and of using a global spatial calibration vs. several regional calibrations of L-VOD – AGB relationships were determined.

The new AGB dataset is then compared against the ESA CCI and Xu et al. (2021) AGB datasets. When using the single global calibration, the L-VOD AGB dataset has lower AGB in equatorial Africa than the ESA CCI AGB dataset, as the logistic fit is not able to reach AGB values over 300 Mg ha-1, which are present in parts of Africa, but not in other densely vegetated regions such as the Amazon rainforest. AGB estimates in equatorial Africa can be improved with the regional calibrations.

General comments:

I believe that the study is of interest to the community and that another above ground biomass dataset covering a longer time period will be useful. The paper is generally well organized, however, in a few sections its clarity could be improved."

Major comments:

The authors should explain the validity and usefulness of comparing an AGB dataset with the Xu et al. (2021) dataset, which combines above ground and below ground biomass more clearly. Why was this dataset chosen, if it covers a different variable?

This dataset was chosen because it is the only available biomass dataset covering a time series longer than 10 years. And it indeed covers a slightly different variable: Xu et al (2021) distribute total (above (AGB) and belowground (BGB)) biomass maps and we created a time series of AGB maps.
In the first version of the manuscript, we directly compared our AGB estimates to the AGB+BGB maps of Xu et al. As suggested by the reviewer, we have reconsidered this comparison for a more relevant one. Indeed, in their supplementary material, Xu et al (2021) provided the list of the root-to-shoot ratios used to convert AGB values to BGB values (Table S5). From Table S5 of Xu et al (2021), we built a simplified root-to-shoot coefficient map in order to convert Xu total live biomass to AGB estimates (see the coefficient map AGB+BGB -> AGB below). We were then able to compare the AGB, i.e. this 'Xu derived AGB', to our L-VOD derived AGB. After this "correction", it appears that the AGB time series trends of Xu et al (2021) are unchanged compared to the AGB+BGB trends. However, the converted Xu et al estimates are closer to our estimates.
We thank the reviewer for pointing out this issue which allows us to establish a more relevant comparison

between both datasets.

[Figure]

Modifications:

*Fig. 9 was updated with 'Xu converted AGB' time series values.*

*Fig 2d and e was updated with the Xu converted AGB values*

*lines 151-153: For this study, the live carbon density maps were converted to above ground carbon density thanks to the root-to-shoot ratios list provided in the supplementary material of (Xu et al., 2021a). The derived above ground biomass (carbon density divided by 0.49 according to Xu et al. (2021a)) map for 2018 reprojected to the EASE 2 grid at 25 km appears on panel (d) of Fig. 2.*

[Figure]

Minor comments:

- l. 29: Clearly define AGB and whether it includes biomass of all vegetation or for example just forests.
We have provided a definition in the introduction.
Modifications: *lines 29-30: The total Above Ground Biomass (AGB, which accounts for the entire land vegetation biomass), is strongly linked to the carbon content in the vegetation (Losi et al., 2003; Djomo et al., 2011).*

- l. 98: Explain what exactly version 700 of the data is.

Version 700 is the version of SMOS products. So, first of all, this sentence was not well-placed, and it was moved to line 90 as: " The SMOS Level 2 (L2) products version 700 (Kerr et al., 2012) between …"

Version 700 corresponds to the version of the algorithm (and the processor) to perform the SM/VOD retrieval. SInce the beginning of the mission, the retrieval chain has evolved. All major changes are associated with a number to keep track of the dataset.  It is important that users specify the version of the data they are using. The details of the evolution are available in this Technical note
https://earth.esa.int/eogateway/documents/20142/37627/SMOS-Level-2-Soil-Moisture-v650-release-note.pdf

and in particular section 2 "Main improvements of the current L2SM version 700 dataset". It specifies the modification of the algorithm, the version of input SMOS brightness temperatures, improved Auxiliary files (that describe the surface).  We think these details are not mandatory for the paper and the objectives of our study.

- l. 104: "located under high" should be "located in the high"
Done.
Modifications: *line 113: These cases are mainly located in the high (>60 ° N) latitudes.*

- Figure 1 caption: "The colored rectangles show the extent of the different regions considered in the study" -> Maybe mention a bit more here about what these regions are or refer to where in the manuscript they are explained.
Done.
Modifications in Figure 1 caption*: (see Sect. 3 for more details about the regions)*

- l. 112: "Santoro and Cartus (2023)" -> Maybe, refer to it as the ESA CCI (or CCI) map like you do below.
Done.
Modifications: *line 121: . The AGB maps from the ESA CCI (Santoro and Cartus, 2023) and Avitabile et al. (2016)*

- l. 125: "resampled to EASE 2 grid" -> Should be "resampled to the EASE 2 grid"
Done.
Modifications: *line 134: resampled to the EASE 2 grid at 25 km.*

- ll. 144-145: Why do you compare the total living biomass from Xu et al. with the CCI AGB? As you're comparing total living biomass with AGB, if the total living biomass exceeds the CCI AGB, it doesn't tell you whether the difference between the two datasets is just in the BGB or the AGB as well.
Please see the reply to the major comment above.

- Figure 2: Why is there no panel b? Why not label panel c as b, even if it is in the second row?
Fig. 2. and references to panels of Fig. 2. have been modified accordingly.

- l. 147: "derive to AGB" -> Remove "to" or replace with "the".
Done.
Modification: line 157: *The methodology used to derive the AGB is detailed in Fig. 3.*

**Responses to community comment:**

Thank you to Boitard for providing the biomass data. However, I have noticed some potential errors in the dataset, as I recently required biomass data for a project. I mapped the net change in AGB from 2010 to 2019 using the data linked in your paper. Surprisingly, it shows that most forest areas globally experienced AGB loss during 2011-2019 (cf PDF). Additionally, I found that the results in your paper differ significantly from the global estimates in Yang et al. (2023), particularly in terms of net change values. Could the authors explain why this discrepancy exists?

Yang, H., Ciais, P., Frappart, F., Li, X., Brandt, M., Fensholt, R., ... & Wigneron, J. P. (2023). Global increase in biomass carbon stock dominated by growth of northern young forests over past decade. Nature Geoscience, 16(10), 886-892.

Thank you for downloading our dataset, analyzing and comparing it to the existing literature. User feedback is needed and precious for us to improve the dataset.

We tried to reproduce the analysis of Yang et al with our data as accurately as possible:
- First, Yang et al. (2023) worked with MgC/ha whereas we work with Mg/ha. The usual rough factor used to convert from biomass to biomass carbon density is 0.5: biomass_carbon_density=biomass*0.5 (Xu et al (2021)). From Yang et al (2023), we were not able to find how the biomass was converted to biomass carbon density. We hence consider the usual 0.5 factor.
- Second, from the Yang et al (2023) article, it was not so clear how Fig 1a was generated. The only information we were able to find lies in the *Total live biomass derived from L-VOD* paragraph: . *The net changes in total live biomass (ΔTB) are calculated as the difference in annual live biomass between 2010 and 2019.* This is not necessarily the most accurate way to compute a trend, especially knowing the quality of SMOS data in 2010 due to : (i) due to intense RFI -Radio Frequency contaminations, and (ii) considering that the January-May 2010 period is not quality checked : see the introduction, page 2 of the following ESA document : https://earth.esa.int/eogateway/documents/20142/37627/SMOS-Level-2-Soil-Moisture-v650-release-note.pdf

We still produced the following map from our AGB estimates with the simple equation:
$\frac{(AGB(2019) - AGB(2011))*0.5}{2019 - 2011 + 1}$. To our understanding, this equation is as close as possible to the data presented in Fig. 1a of Yang et al (2023). The three maps (our estimates, the one produced by the commenter and the one from Yang et al (2023)) are shown below for comparison:

a) Our estimate of Net Change 2019 - 2011 (MgC/ha/yr)

[Figure]

b) Net change during 2011-2019  Xiaojun Li based on our dataset

[Figure]

Net change (MgC ha$^{-1}$ yr$^{-1}$)

c) Total live biomass changes  2019-2010 Yang et al (2023)

[Figure]

Overall, it seems to us that the global spatial patterns of net change are similar between our AGB estimates and the Yang et al (2023) article. We indeed get more extreme values than Yang et al (2023) but otherwise over most areas the changes present the same trends (except some AGB loss areas in Alaska and North Western Russia).

The observed differences in change magnitude and AGB loss/gain is some areas can be explained by the differences in input data and methodology used between Yang et al and us:
- Yang et al is total live biomass (AGB+BGB), ours is representative of AGB only;
- Yang et al took into account the 'legacy' $CO_2$ emissions from decaying coarse woody debris (CWD); We did not take such a component into account;
- Yang et al's year of reference to calibrate the AGB-L-VOD spatial relationship is 2010. We took 2018;
- Yang et al used 4 different AGB reference maps (ref 52 to 55 in their article) we only used one;
- Yang et al established 13 sets of calibration parameters from 2 calibration functions (Table S1 from their article). We used a single calibration for the whole Earth even though we studied the impact of regional calibration. We found this impact to be small except for the African continent;
- Yang et al use SMOS-IC data, we use SMOS-L2 and the data filtering is not exactly the same;

Finally, as also stated by Yang et al (2023): *All the existing AGB maps contain uncertainties and bias, and none can be considered as closer to the truth (true AGB map is unknown).* Hence, it is fundamental to share with the community different datasets computed from different input data and processed using different approaches, to compare them with each other.

**Responses to supplementary comment of topic editor:**

The justifications provided in the Introduction about the carbon cycle, carbon sinks, or sources are still confusing since Aboveground biomass is not the same as Aboveground carbon biomass (Aboveground carbon biomass = Aboveground biomass x carbon_content). Please check and revise for clarity.

We have modified the first paragraph of the introduction to make it clearer.
Modifications:
*line 24: In particular, the forest carbon biomass constitutes a large carbon reservoir as [...].*
*line 29-31: The total Above Ground Biomass (AGB, which accounts for the entire land vegetation biomass), is strongly linked to the carbon content in the vegetation (Losi et al., 2003; Djomo et al., 2011). Hence, mapping this essential climate variable and following its evolution in space and time is fundamental to a precise and reliable monitoring of the Earth carbon balance.*

---

## Referee Report (RR1)

Comments for Manuscript: Earth System Science Data
Article ID: essd-2024-184
Title: Above ground biomass dataset from SMOS L band vegetation optical depth and reference maps

In the manuscript, to improve the accuracy of the AGB estimates, the authors produce a harmonized AGB dataset from the L-VOD and analyze the impact of different factors on the AGB/VOD calibration. Global maps derived from SMOS L-VOD and reference AGB maps are thought to be of fundamental importance to address the uncertainty in the estimates of AGB spatial distribution and temporal evolution. Averaging the SMOS L-VOD over a year and linking it to a pre-existing AGB map is a well-established method to derive a spatial relationship between both quantities. After temporal extrapolation of this relation, global AGB time series are derived from the L-VOD, allowing to retrieve vegetation biomass values up to 300 Mg ha$^{-1}$ from 2011 onwards. Overall, the paper is very well written and scientifically rigorous.

My only recommendation is that an additional analysis/discussion be provided showing the uncertainties of input datasets, methods, and final results.

Here are my concerns about this manuscript:
1. The section of Abstract is too lengthy now, which should be written in a pithy style, with some specific quantitative results.

2. The logical structure of the manuscript is somewhat confusing, such as the methods and results section. It is recommended to consult an expert for further polishing.

3. In Discussion section, please give evidence to support these findings in the manuscript.

4. In Conclusion section, the cited references should be deleted. Meanwhile, give some specific numbers to conclude the research results.

---

## Author Response (AR2)

**General modifications:**

We took the opportunity of this second round of reviews to update our paper with the latest released version of the CCI Biomass maps. The version 5.01 of the CCI was released in July 2024 and distributes global biomass maps for 2010 and 2015-2021. Besides the usual 100m resolution maps, aggregated maps at 1 km, 10 km, 25 km and 50 km are distributed and we are now working with the 25 km aggregated maps. Using v5 of the CCI rather than v4 also ensures a longer (2015-2021 rather than 2017-2020) and a state-of-the-art time series comparison to contextualize the results presented in our article without changing its general conclusions. The philosophy, principle and main results are not altered by this update. The statistics have been updated with the new values but beyond the values of the results, their interpretation remains the same.

Moreover, we have slightly strengthened the filtering of our input VOD. On top of the Chi2P<0.05 or RFI>20% filtering we have set up the following:

- The footprints which median RFI value over the year is above 20% are entirely dropped for that year
- The footprints for which there are fewer than 10 available quality measures over the year are entirely discarded for that year
- The VOD acquired when the air temperature is below 273K are also discarded.

This more restrictive filtering increases the quality of the input VOD but does not change the philosophy, principle and main results of the article. More pixels are discarded in the process but the remaining ones are of higher quality. The time series of estimated AGB maps are not significantly changed by this update. All of the above-mentioned changes are described in the revised version of the paper.

**Responses to Anonymous Referee #1**

We thank referee #1 for providing a second feedback on our revised manuscript. The valuable feedback was carefully taken into account to improve the paper.

Below, Referee #1 comments are marked in red.
Responses to the comments are marked in blue.
Changes made in the manuscript are marked in *italic*.

Minor comments:

Thanks a lot to the authors for making the effort to review the manuscript! Most of my concerns were addressed. However, I still believe that the following issues should be clarified:

1. Regarding the aggregation of the uncertainty map by using a quadratic mean of all non-nodata contributing pixels - this automatically reduces the uncertainty estimate obtained for the SMOS resolution. The uncertainty of an average is usually lower than the uncertainty of the individual pixels. How much the uncertainty is reduced depends on how strongly the errors are correlated between pixels. I understand that the authors did not really take this into consideration, so I think that you should clearly explain this in the manuscript.

Indeed, thank you for pointing it out. Following this round of review, we have decided to update our manuscript with the just-released version 5 of the Biomass CCI. In the latest version, the CCI AGB maps and their associated uncertainties are distributed at several ground resolutions (100 m and aggregated at 1 km, 10 km, 25 km, 50 km). We then directly worked with the aggregated resolution of 25km. We added a sentence to clarify this, and also to specify that the aggregated uncertainty is lower than the original uncertainty.
Modifications:

*lines 232-234: The standard deviation of the aggregated maps at 25 km is significantly lower than the standard deviation of the original resolution (100 m) maps and lies below 15% of the AGB value for most pixels*

2. I completely agree that validating AGB or L-VOD datasets in the temporal domain is tricky and that one cannot expect agreement with optical data in most of the biomes. However, given that this dataset is one of the only AGB datasets with multiple time steps, I think it would be important to include time series of your AGB estimate compared to other AGB estimates (Xu).

We agree with this comment, and this is the reason why such comparisons are included in the paper, see Figures 8 and 9. As there is no benchmark AGB map, the time series comparison was performed with the two datasets available over multiple years: Santoro et al. (CCI biomass) and Xu et al.

**Responses to Anonymous Referee #3**

We thank the referee #3 for reviewing and commenting on the manuscript. This valuable feedback was carefully taken into account to improve the quality of the paper. It seems like a new review to us as such concerns were not raised during the first round. This is why they were not previously addressed.

Below, Referee #2 comments are marked in red.
Responses to the comments are marked in blue.
Changes that have been made in the manuscript are marked in *italic*.

This study derives yearly Above ground biomass (AGB) from the vegetation optical depth of SMOS, taking advantage of the L band of SMOS. This leads to a new set of data readily available for use in scientific research and other potential areas.
The product is demonstrated to be reliable, and its uncertainties are assessed to properly understand its performance.
There is a lot of clarity about how the data is developed, and the study also aims at answering some questions that lead to decisions made in the development of the product. I do have some concerns that I hope will make some issues clearer and perhaps, the data more helpful to users.

General comments
1.      The two hypotheses chosen here are nice but might need proper reframing. The first one, looking at the impact of orbit time is already well known. Additionally, I don't see it's relevance at the timescale the AGB product is developed. Finer timescales will benefit more from this. Furthermore, it doesn't really inform/optimize the Asc+Desc merging routine, to perhaps, provide weights to pixels or time points of higher qualities. At this time scale, I would be very surprised if the difference between the uncertainties of the orbit times are anything worth taking seriously. Studies of Thomas Holmes (NASA Goddard), for instance, have shown that the SMOS overpass times are the cremes for thermal equilibrium. I would be more interested in this hypothesis if one of the overpass times were somewhere close to something like AMSR2's, close to midday. I think a proper reframing may be required.

Indeed, the SMOS orbital configuration was set up at 6am specially for the mentioned reasons (before T. Holmes' studies). By construction, the counterpart of the orbit is 6:00 PM. The reviewer is correct to say that more differences are expected between the orbits (Ascending vs Descending) for daily/season variability studies (see 3rd and 4th images below for a comparison of the yearly and monthly difference between Asc. and Desc. orbits). Ideally, we would only use ascending orbits if we could. Unfortunately, RFI (Radiometric Frequency Interference) prevents SMOS from acquiring data everywhere (China, Eastern Siberia, Africa may be impacted). Ascending (A) and descending (D) orbits are impacted differently by the RFI (see the first two

example images below for 2018) and including descending orbits in the processing leads to an increased number of observations and an extended coverage of the Earth. As shown by the third figure below, ascending and descending yearly averaged VOD still present non negligible differences even though these differences are smoother than when averaging over a month (4th and last image). According to us, testing how the calibration may be impacted by merging both orbit types remains relevant but, as you suggested, needs to be specified and reframed.

Indeed, our hypothesis is more about checking that both ascending and descending orbits can be merged properly rather than checking whether an orbit type is better than the other (we know that ascending orbits are better by design for this mission). When conducting this work, we were expecting that including descending orbits would not impact the estimation significantly but we still wanted to check it out, especially in the areas where ascending orbits are not available due to RFI. With this test, we specifically want to demonstrate that both orbits (AM and PM) can be merged properly without great consequences on the calibration.

Minimum, Median, Maximum RFI and Number of valid points for the ascending (A) orbits of 2018:

[Figure]

Minimum, Median, Maximum RFI and Number of valid points for the descending (D) orbits of 2018:

[Figure]

Min RFI 2018 Orbit D

Med RFI 2018 Orbit D

Max RFI 2018 Orbit D

NValid Points 2018 Orbit D

Difference of the yearly averaged VOD: Asc - Desc Orbits 2018

[Figure]

**Difference of the monthly averaged VOD: Asc - Desc Orbits July 2018**

Modifications: The hypothesis has been specified along the article:

*Abstract line 10-11: First, the spatial calibrations obtained with VOD from different orbit types are compared to check whether ascending and descending VOD can properly be merged.*

*Introduction line 74-75: The work carefully investigates the influence of: mixing morning and afternoon overpasses to maximize the number of observations per node*

*line 163: The first factor is the relevance of merging the SMOS overpasses with different local time*

*lines 166-168: Nevertheless, descending orbits will help fill the spatial gap in areas strongly affected by RFI. If both orbit types can be merged properly without impacting the quality of the calibration, the spatial extent of the AGB estimations will be increased.*

*Title of section 4.2 line 267: Relevance of merging ascending and descending orbits*

*line 276: Considering these results, ascending and descending orbits can properly be merged to compute the yearly L-VOD maps.*

*Sparse modifications in the paragraph lines 357 to 364: The effect of the time of observation is also evaluated. It is admitted that morning overpasses (6 am for SMOS) offer more stable surface conditions as the Earth's surface reaches a thermal equilibrium. Therefore, better (SM, VOD) retrievals are expected using the morning orbits. It has however no impact for the present application as the three cases (i.e. only morning, only afternoon, and the two combined) have similar performances at the timescale considered in this study, with a correlation coefficient R ranging between 0.85 and 0.87 (see Table1). The yearly averaging smooths the daily variability, caused by the vegetation water content. Both overpasses (morning and afternoon) can be merged without a strong impact on the calibration. This increases the number of observations per pixel to compute the yearly VOD average and improves the coverage of regions polluted with RFI.*

*lines 420-421: In particular, it is shown that SMOS ascending and descending orbits can and should be merged at the yearly time scale to estimate the AGB.*

2.     The second hypothesis of regional optimization, I think is a very nice one, but I am also surprised about the regional demarcations chosen in this study. I don't see the authors using lessons learned from VOD in previous studies to inform the choice of the selected regions. I provide more details below to hopefully help with this.

Lines 352-362: I think there might be other ways of properly understand the uncertainties in the datasets. just choosing a region like Africa is not a very practical choice since the continent has at least, three significant climate conditions that would impact the uncertainty propagation in the region. Choosing to look at how the metrics vary over VOD scenarios(perhaps binned VOD values from about 0.15 to 0.9) is a probably a better way to understand uncertainty propagations into the AGB product.

We grouped the reviewer's general comment 2 with their other comment 3 as these two comments point out the same concern on the chosen regional demarcations (specifically the African continent).

The African continent is indeed a vast region that encompasses several climatic conditions. However, the choice of this specific area is based on previous VOD-AGB studies (Rodriguez-Fernandez et al. 2018, Mialon et al. 2020, Wang et al. 2024). The entire African continent was also chosen as the L-VOD is very well correlated to the whole ecosystems of this continent (Wang et al. 2024, Rodriguez-Fernandez et al. 2018, Brandt et al. 2018 ), so we thought it would be useful for analysis and comparisons with these published works.

When designing the study, over Africa, we first conducted our analysis based on the IGPB landcover classification (as in Mialon et al. 2020). We established linear relationships between the L-VOD and the AGB per land cover class (see first image below). This naturally creates different VOD scenarios as suggested by the reviewer. Then, we estimated the AGB from the L-VOD per IGBP land cover over Africa. As shown by the second figure below, this gives very close results to the regional calibration with a single logistic fit. Over Africa, estimating the AGB per IGBP class does not impact the estimation as much as Global vs Regional estimates (Fig 7a)

Linear regressions between L-VOD and AGB per IGBP class over Africa - 2018:

[Figure]

Histogram of the differences over Africa between the calibration AGB from CCI v5 and the estimated AGB from a regional logistic fit (grey) and several linear fits per IGBP class (black):

[Figure]

As for the choice of the other regions (Tropics, Amazon and boreal regions), after conducting a per IGBP class analysis for the whole Earth, we figured that some classes could be grouped because they showed very similar results. Then, exchanges with biomass experts led us to consider and to analyze more particularly the forested regions such as the boreal and the tropical forests.

Rodríguez-Fernández, N. J., Mialon, A., Mermoz, S., Bouvet, A., Richaume, P., Al Bitar, A., Al-Yaari, A., Brandt, M., Kaminski, T., Le Toan, T., Kerr, Y. H., and Wigneron, J.-P.: An evaluation of SMOS L-band vegetation optical depth (L-VOD) data sets: high sensitivity of L-VOD to above-ground biomass in Africa, Biogeosciences, 15, 4627–4645, https://doi.org/10.5194/bg-15-4627-2018, 2018.

Mialon, A.; Rodríguez-Fernández, N.J.; Santoro, M.; Saatchi, S.; Mermoz, S.; Bousquet, E.; Kerr, Y.H. Evaluation of the Sensitivity of SMOS L-VOD to Forest Above-Ground Biomass at Global Scale. *Remote Sens.* **2020**, *12*, 1450. https://doi.org/10.3390/rs12091450

Mengjia Wang, Philippe Ciais, Rasmus Fensholt, Martin Brandt, Shengli Tao, Wei Li, Lei Fan, Frédéric Frappart, Rui Sun, Xiaojun Li, Xiangzhuo Liu, Huan Wang, Tianxiang Cui, Zanpin Xing, Zhe Zhao, Jean-Pierre Wigneron, Satellite observed aboveground carbon dynamics in Africa during 2003–2021, Remote Sensing of Environment, 2024, https://doi.org/10.1016/j.rse.2023.113927

Brandt, M., Wigneron, J.-P., Chave, J., Tagesson, T., Penuelas, J., Ciais, P., … Fensholt, R. (2018). Satellite passive microwaves reveal recent climate-induced carbon losses in African drylands. Nature Ecology & Evolution, 2(5), 827–835. doi:10.1038/s41559-018-0530-6

3.      Thirdly, I am sure the authors are already aware that we don't really have any standard data for benchmarking AGB at such a global scale, thus, using any statistical analysis to quantify uncertainties needs to be done with care, especially when we use words like errors, which imply there is a benchmark. Perhaps, RMSD sounds like a more thoughtful word instead of RMSE. Biases basically inform us of which product may be potentially dryer or wetter and in which regions. And this interpretation should be more pronounced in the paper. In this case, Biases don't necessarily mean deviations as there is no standard.

Indeed, we fully agree with the reviewers comments. First of all, we changed the statistics and replaced the RMSE with the unbiased RMSD (as suggested in your other comment #2). The maps we used for calibration are our references for the study but are no benchmark.  The reviewer also pointed out the wrong use of "error". We also agreed that we should use "uncertainties", and changed the manuscript accordingly.

Modifications:

line 274: *For this study, the bias is not the most important metric as it does not necessarily reflect a deviation from the true AGB value which is unknown, as no benchmark AGB map exists.*

Other Comments

1.      Line 55: Perhaps it is more helpful to properly introduce the L-band VOD product here. The current form makes it look quite trivial.

We changed the sentences, to hopefully better introduce the VOD.

Modifications

*line 56 : " This L-band VOD (or L-VOD, Wigneron et al., 2007) characterizes how much the surface signal is attenuated by the vegetation, as well as the vegetation's own emission. It is then defined as an optical depth. Theory and previous works (Jackson and Schmugge, 1991; Grant et al.,2012), showed that the L-band VOD is strongly influenced by the Vegetation Water Content (VWC). "*

2.      Lines 215-218: Authors should note that these statistics are no independent. Therefore, one would hardly find additional information from RMSE or Bias if R is already known. Secondly, I disagree with the choice of RMSE and Bias since there are not blind to systematic errors that may mask out relevant information from the anomalies of the data. In such a case, R becomes the most reliable metric or some sort of unbiased RMSD?

We agree that the RMSE and the bias are not independent. As suggested, we replaced the RMSE by its unbiased counterpart that is the ubRMSD, with the "D" referring to a difference (and not an error).  We think the bias can add information for our analysis (e.g. regional vs global relationships), even though R and the ubRMSD are more important for our study. As shown in Fig 7, similar R values can hide very different ubRMSD and biases (example of the African continent).